# Efficient Bayesian Inference from Noisy Pairwise Comparisons

**Till Aczel** [1 2]  **Lucas Theis** [2]  **Roger Wattenhofer** [1]

## Abstract

Evaluating generative models is challenging because standard metrics often fail to reflect human preferences. Human evaluations are more reliable but costly and noisy, as participants vary in expertise, attention, and diligence. Pairwise comparisons improve consistency, yet aggregating them into overall quality scores requires careful modeling. Bradley-Terry-based methods update item scores from comparisons, but existing approaches either ignore rater variability or lack convergence guarantees, limiting robustness and interpretability. We introduce BBQ, a Bayesian Bradley-Terry variant that explicitly models rater quality, downweighting or removing unreliable participants, and provides guaranteed monotonic likelihood convergence through an Expectation-Maximization algorithm. Empirical results show that BBQ provides efficient inference, well-calibrated uncertainty estimates, and more robust, interpretable rankings compared to baseline Bradley-Terry models, even with noisy or crowdsourced raters. This framework enables more reliable and cost-effective human evaluation of generative models.

## 1. Introduction

Evaluating generative models is challenging, particularly for large language models (LLMs) and image generators, where standard metrics often fail to reflect human preferences. Metrics such as BLEU (Papineni et al., 2002) and perplexity (Jelinek, 1998) for LLMs, or PSNR (Gonzalez, 2009), MS-SSIM (Wang et al., 2003), and FID (Heusel et al., 2017) for image models, provide only limited insight into perceived quality (Mentzer et al., 2020; CLIC, 2025; Chiang et al., 2024). As a result, human evaluations remain indispensable for establishing meaningful rankings between

[1]ETH Zurich, Switzerland [2]Mabyduck, United Kingdom. Correspondence to: Till Aczel <taczel@ethz.ch>.

*Proceedings of the 43rd International Conference on Machine Learning*, Seoul, South Korea. PMLR 306, 2026. Copyright 2026 by the author(s).

models, motivating the widespread use of human preference benchmarks in the evaluation of large language models and Learned Image Compression (LIC) methods.

However, human evaluations are expensive and time-consuming, making it crucial to design protocols that are efficient while minimizing subjectivity and noise. In this context, pairwise comparisons, in which participants choose between two items rather than providing absolute scores, represent an effective and practical form of human evaluation (Zerman et al., 2018; Wang et al., 2023). They are generally easier for participants and produce more consistent judgments. Collecting all pairwise comparisons is infeasible because the number of pairs grows quadratically with the number of items. With only a limited set of comparisons, simple statistics such as win rates are not sufficient to derive overall quality scores, because a win rate depends on the quality of the items it is compared against.

To produce an overall ranking that can be displayed on leaderboards, pairwise comparisons must be aggregated effectively. This motivates the development of robust and efficient aggregation methods. The Bradley-Terry (BT) (Bradley & Terry, 1952) model addresses this problem by iteratively updating item scores based on comparison outcomes. It has become a standard approach for aggregating pairwise judgments across domains.

Despite its usefulness, the standard Bradley–Terry model has several limitations in practice. There are multiple algorithms for estimating BT parameters, including iterative scaling methods (Ford Jr, 1957), gradient descent, and the minorization maximization (MM) algorithm (Hunter, 2004). The MM algorithm guarantees convergence to the global maximum for the basic BT model, whereas gradient-based methods, which are closely related to the Elo rating system, are widely used in practice but provide no such guarantees (Hunter, 2004). For extended BT models that incorporate factors such as home field advantage (Agresti, 2010), multiple comparisons (Plackett, 1975; Luce et al., 1959), or ties (Rao & Kupper, 1967), even MM algorithms may converge only to local optima (Hunter, 2004). A further challenge is that human evaluations are inherently noisy: participants may rush, guess, or lose focus, leading to uneven reliability across raters. If such variability is ignored, unreliable raters can distort item scores and reduce ranking stability.

We propose a Bayesian Bradley–Terry variant that jointly models item quality and rater reliability. To the best of our knowledge, no prior work combines the BT framework with a Bayesian formulation that explicitly models rater quality and provides closed-form EM updates. Previous approaches that account for noisy raters typically rely on gradient-based optimization without convergence guarantees, whereas our method ensures stable convergence and yields interpretable parameters. The Bayesian framework addresses epistemic uncertainty, provides regularization, and ensures stable convergence of the estimation procedure. To capture variability in participant behavior, our method introduces rater-specific parameters that reflect how consistent or trustworthy each participant is, allowing the model to adjust the influence of individual comparisons. We derive an EM algorithm that efficiently estimates item and rater parameters via a latent-variable formulation, guaranteeing monotonic likelihood improvement. This approach allows partially reliable raters to contribute meaningful information while reducing the impact of inattentive or inconsistent participants. The Bayesian priors further regularize the estimates, preventing overfitting when many raters and parameters are involved, and leading to stable and generalizable score estimates. By modeling rater quality and adopting a Bayesian framework, our method also provides uncertainty estimates for item scores, facilitating more interpretable rankings and robust comparisons across studies.

We demonstrate the effectiveness of our approach on human evaluation datasets for generative models, showing efficient inference, improved robustness to noisy comparisons, and more consistent rankings compared to standard BT and naive aggregation methods. Beyond generative model evaluation, our framework can be applied to any scenario where noisy pairwise comparisons must be converted into reliable global rankings. Overall, our work contributes to more cost-efficient, interpretable, and reproducible human studies for evaluating AI-generated content.

**Conflict of Interest Disclosure.** This research was conducted while Till Aczel and Lucas Theis were employed by Mabyduck Ltd., whose services involve ranking and rater-quality evaluation based on Bradley-Terry models as studied in this paper. Mabyduck funded the user study that collected the IHQ dataset analyzed here.

## 2. Related Work

Human studies often require ranking items. Presenting participants with a choice between two items rather than a single item with a score increases sensitivity to subtle differences and reduces variability in responses (Zerman et al., 2018; Wang et al., 2023). Consequently, many leaderboards for machine learning models rely on pairwise comparisons.

In such setups, participants indicate their preference between two alternatives, and these preferences are then aggregated to produce overall rankings. Notable examples include the Chatbot Arena (Chiang et al., 2024) and the CLIC image compression challenge (CLIC, 2025), which use pairwise comparisons and combine them using variants of the Bradley-Terry (BT) model.

The Bradley-Terry model (Bradley & Terry, 1952) was originally developed to rank competitors in sports. It provides a probabilistic framework to estimate the likelihood that one item is preferred over another. The model converts pairwise comparisons into a ranking, making it a cornerstone in studies across games, consumer preferences, and other applications. Several extensions have been proposed to broaden its applicability. For instance, the Plackett-Luce model generalizes the BT framework from pairwise comparisons to rankings over multiple items (Plackett, 1975; Luce et al., 1959), defining a probability distribution over permutations by multiplying successive BT probabilities (Luce et al., 1959). Other modifications address specific contexts, such as modeling home-field advantages (Agresti, 2010), incorporating ties (Rao & Kupper, 1967), or handling comparisons between groups instead of individuals (Huang et al., 2006).

Beyond Bayesian BT variants, the pairwise-comparison literature also studies efficient recovery of top items or full rankings under fixed comparison budgets. Examples include spectral and comparison-graph approaches such as Rank Centrality (Oh, 2017), as well as methods targeting robust or approximate ranking from pairwise data (Shah & Wainwright, 2018; Heckel et al., 2018). These methods focus on ranking recovery from pairwise outcomes, typically under homogeneous-rater assumptions, whereas our setting emphasizes Bayesian aggregation with explicitly heterogeneous rater quality.

Maximum likelihood estimation (MLE) is commonly used to infer the parameters of the basic BT model. Zermelo (1929) introduced an iterative approach to compute these estimates, which has become widely adopted. Later, Lange et al. (2000) demonstrated that this procedure is a particular case of minorization-maximization (MM) algorithms, which iteratively optimize surrogate functions to reach a local maximum of the likelihood. Hunter (2004) extended MM algorithms to generalized BT models and established conditions guaranteeing convergence to the MLE. For the classical BT model, the optimization is convex, ensuring convergence to the global optimum (Hunter, 2004). However, for extensions such as home-field advantage, multiple comparisons, or ties, MM algorithms may converge only to a local maximum (Hunter, 2004).

Bayesian formulations of the BT model have been explored to incorporate prior knowledge and regularization (Adams,

2005; Guiver & Snelson, 2009; Caron & Doucet, 2012). Caron & Doucet (2012) showed that MM algorithms can be interpreted as Expectation-Maximization (EM) procedures, enabling Bayesian inference via Gibbs sampling. This formulation provides tractable complete-data likelihoods and ensures convergence of the resulting Markov Chain Monte Carlo methods. The Bayesian perspective smooths posterior estimates, reducing susceptibility to local maxima, and allows for uncertainty quantification in the estimated item skills.

Annotator quality models represent a critical extension addressing the assumption that all comparisons are equally reliable. In crowdsourced evaluations, participant quality can vary widely, motivating approaches that explicitly account for annotator reliability (Chen et al., 2013). Chen et al. (2013) proposed a Bayesian model that jointly estimates both item quality and rater reliability using annotator-specific parameters. Their approach places Gaussian priors on item and rater parameters, and incorporates scaling factors in the likelihood to modulate individual annotator influence. They perform posterior inference using gradient-based optimization, which can be challenging in high-dimensional spaces due to potential local optima. Moreover, gradient-based methods have no guarantees of convergence. In contrast, our proposed EM-based method provides guaranteed monotonic likelihood improvement. The Elo-based rater model developed by Google Research implements a simplified version of this approach. It is widely used in the Challenge on Learned Image Compression (CLIC) (CLIC, 2025) and related research (Mentzer et al., 2020; Ballé et al., 2025).

To the best of our knowledge, no prior work combines the Bradley–Terry model with a Bayesian formulation that explicitly models rater quality while also providing closed-form EM updates. In contrast to Chen et al. (2013), who use Gaussian priors and gradient-based optimization without convergence guarantees, our approach leverages conjugate priors and an EM algorithm that ensures monotonic likelihood improvement. Compared to the Elo-based rater model widely used in CLIC (CLIC, 2025), which is a simplified heuristic relying on Elo-style updates, our method provides a principled Bayesian treatment with uncertainty estimates and interpretable rater-quality parameters.

## 3. Methodology

### 3.1. Bradley-Terry model with rater quality

The objective of converting a set of noisy pairwise comparisons into a reliable ranking of items is a fundamental problem in machine learning and statistics. Consider a set of $K$ items that are repeatedly compared with one another in pairs by a set of $R$ raters. The data, which we denote as

$D$, consists of the outcomes of these comparisons. For two items $i$ and $j$ of this set, Bradley & Terry (1952) suggested the following model:

$$P(i \text{ beats } j) = \frac{\lambda_i}{\lambda_i + \lambda_j} \quad (1)$$

where $\lambda_k > 0$ is a parameter associated with item $k \in \{1, 2, \ldots, K\}$ that represents its skill rating. This model provides a clear and interpretable way to infer item scores from a set of observed wins and losses. However, it operates under the simplifying assumption that all comparisons are equally reliable. In the context of human evaluation, this assumption is often violated. Participants may exhibit varying levels of expertise, attention, or diligence, leading to unreliable and inconsistent judgments.

To account for varying rater reliability, we introduce a rater-specific quality parameter $q_r \in [0, 1]$. Intuitively, with probability $q_r$, rater $r$ makes an informed judgment following the Bradley-Terry model. With probability $1 - q_r$, the rater guesses randomly, as if flipping a fair coin between the two items. This leads to the following mixture model for the probability that rater $r$ ranks item $i$ above item $j$:

$$P(r \text{ ranks } i \text{ above } j) = q_r \left( \frac{\lambda_i}{\lambda_i + \lambda_j} \right) + (1 - q_r) \left( \frac{1}{2} \right) \quad (2)$$

The log-likelihood of the data is given by:

$$\log P(D \mid \lambda, q) = \\ \sum_{r=1}^{R} \sum_{i=1}^{K} \sum_{\substack{j=1 \\ j \neq i}}^{K} \left[ w_{r,ij} \log \left( q_r \frac{\lambda_i}{\lambda_i + \lambda_j} + (1 - q_r) \frac{1}{2} \right) \right. \\ \left. + (n_{r,ij} - w_{r,ij}) \log \left( q_r \frac{\lambda_j}{\lambda_i + \lambda_j} + (1 - q_r) \frac{1}{2} \right) \right], \quad (3)$$

where $n_{r,ij}$ denotes the total number of comparisons between items $i$ and $j$ by rater $r$, and $w_{r,ij}$ is the number of times rater $r$ ranks item $i$ above item $j$. There is no closed-form maximum likelihood estimator for this likelihood function, so the optimal parameters $\lambda$ and $q$ cannot be derived analytically. While an iterative optimization method like gradient descent could be used, it offers no guarantee of convergence. On the other hand, the Expectation–Maximization (EM) algorithm avoids learning-rate tuning, and guarantees a monotonic increase of the observed-data likelihood and convergence to a stationary point (Dempster et al., 1977).

### 3.2. Thurstonian Interpretation

Following Caron & Doucet (2012), we interpret the Bradley-Terry model under a Thurstonian framework (Diaconis,

1988). In this perspective, a comparison between items $i$ and $j$ is conceptualized as a race, where each item has a random arrival time, $Y_i$ and $Y_j$, respectively. These arrival times are assumed to follow exponential distributions:

$$Y_i \sim \mathcal{E}(\lambda_i), \quad Y_j \sim \mathcal{E}(\lambda_j), \quad (4)$$

and the item with the smaller arrival time is declared the winner. This leads directly to the standard Bradley-Terry probability:

$$P(i \text{ beats } j) = P(Y_i < Y_j) = \frac{\lambda_i}{\lambda_i + \lambda_j}. \quad (5)$$

For the EM algorithm, we introduce latent variables to simplify the complete-data likelihood. First, we define an indicator variable

$$A_{r,ij}^{(c)} \sim \text{Bernoulli}(q_r), \quad (6)$$

which denotes whether the $c$-th comparison of items $i$ and $j$ by rater $r$ follows the Bradley-Terry model. Using these indicators, we define the latent variable $Z_{r,ij}$ as the sum of the minimal arrival times across the $n_{r,ij}$ comparisons by rater $r$:

$$Z_{r,ij} = \sum_{c=1}^{n_{r,ij}} A_{r,ij}^{(c)} \, \min(Y_i^{(c)}, Y_j^{(c)}). \quad (7)$$

Conditioned on $m_{r,ij} = \sum_{c=1}^{n_{r,ij}} A_{r,ij}^{(c)}$, the variable $Z_{r,ij}$ follows a Gamma distribution,

$$Z_{r,ij} \mid m_{r,ij} \sim \Gamma\big(m_{r,ij}, \lambda_i + \lambda_j\big), \quad (8)$$

where $\Gamma(\alpha, \beta)$ denotes the Gamma distribution with shape parameter $\alpha$ and inverse scale $\beta$. These latent variables are introduced primarily for conjugacy and closed-form EM updates, following the auxiliary-variable construction of Caron & Doucet (2012). This Gamma-distributed latent variable formulation therefore yields a tractable EM update while accounting for rater-specific quality.

### 3.3. Expectation-Maximization Updates

The EM algorithm is an iterative method for finding maximum a posteriori (MAP) estimates for our model parameters, $\lambda$ and $q$, by treating the rater's quality and the unobserved arrival times from the Thurstonian interpretation as latent variables. The algorithm is guaranteed to converge to a stationary point of the posterior distribution.

First, we specify prior distributions for the parameters. The item skills $\lambda_k$ are assigned a Gamma prior, $\lambda_k \sim \Gamma(a, b)$, which is conjugate to the exponential-race construction. Each rater's quality parameter, $q_r$, is given a Beta prior, $q_r \sim B(\alpha, \beta)$. These conjugate choices keep inference

stable and efficient by yielding closed-form EM updates; Section C reports sensitivity ablations over these hyperparameters.

The core of the EM algorithm is the iterative maximization of the expected complete-data log-posterior, conditioned on the current parameter estimates $(\lambda^*, q^*)$. The algorithm proceeds by alternating between two steps until convergence:

**E-step: Expectation** In the E-step, we compute the expected complete-data log-posterior, a function we denote as $Q$. This function represents the expected value of the log-posterior of all observed and latent variables, given our observed data and the current parameter estimates from the previous iteration. It is defined as:

$$Q(\lambda, q \mid \lambda^*, q^*) = \mathbb{E}_{A,Z|D,\lambda^*,q^*}\Big[\ell_c(\lambda, q; D, Z, A) \\ + \log P(\lambda) + \log P(q)\Big]. \quad (9)$$

The complete-data log-likelihood, $\ell_c$, is further broken down into three components.

$$\ell_c(\lambda, q; D, Z, A) = \log P(Z \mid D, A, \lambda, q) \\ + \log P(D \mid A, \lambda, q) \quad (10) \\ + \log P(A \mid \lambda, q).$$

By conditional independence, these terms simplify to $\log P(Z \mid A, \lambda)$, $\log P(D \mid A, \lambda)$, and $\log P(A \mid q)$; the full derivation is given in Appendix A.

**M-step: Maximization** This step updates the model parameters by maximizing the $Q$ function. By leveraging the expected values from the E-step, the M-step transforms the original complex optimization problem into simpler, closed-form updates. The key quantity that is used in both updates is the posterior probability that a given comparison from rater $r$ follows the Bradley-Terry model. This quantity, denoted as $\gamma$, represents the weight of a rater's judgment based on how much it aligns with the model's current predictions. It is given by:

$$\gamma_{r,ij}^{(t-1)} = \frac{q_r^{(t-1)} y_{ij}^{(t-1)}}{q_r^{(t-1)} y_{ij}^{(t-1)} + (1 - q_r^{(t-1)}) \frac{1}{2}}, \quad (11)$$

where $y_{ij}^{(t-1)} = \frac{\lambda_i^{(t-1)}}{\lambda_i^{(t-1)} + \lambda_j^{(t-1)}}$ is the Bradley-Terry probability that item $i$ beats item $j$. This posterior probability $\gamma$ represents our confidence that a comparison was meaningful rather than random, given the current parameter estimates. Higher $\gamma$ values indicate more trustworthy comparisons.

The new estimate for a rater's quality, $q_r$, is calculated as a weighted average. The numerator sums up the "effective number of wins" for that rater, where each win is weighted

by the posterior probability ($\gamma$) that it was a meaningful, non-random judgment. This is combined with the hyperparameters from the Beta prior to regularizing the estimate. The denominator normalizes this sum by the total number of comparisons and prior parameters. This update intuitively increases a rater's quality score if their judgments frequently align with the model's predictions. The update is given by:

$$q_r^{(t)} = \frac{\sum_{i=1}^{K}\sum_{j=i+1}^{K}\left[w_{r,ij}\,\gamma_{r,ij}^{(t-1)} + w_{r,ji}\,\gamma_{r,ji}^{(t-1)}\right] + (\alpha - 1)}{n_r + \alpha + \beta - 2}, \tag{12}$$

where $n_r$ is the total number of comparisons by rater $r$.

The new estimate for an item's skill, $\lambda_i$, is a ratio that balances two key quantities. The numerator is a sum of the "effective wins" for item $i$ across all raters, where each win is again weighted by the rater's quality ($\gamma$). This term essentially represents the total positive evidence for item $i$. The denominator, on the other hand, accounts for the total comparisons item $i$ was involved in, and acts as a normalizing factor. These terms are also regularized by the Gamma prior hyperparameters. The update is given by:

$$\lambda_i^{(t)} = \frac{\sum_{r=1}^{R}\left[\sum_{j=1,j\neq i}^{K} w_{r,ij}\,\gamma_{r,ij}^{(t-1)}\right] + (a-1)}{\sum_{j=1,j\neq i}^{K}\left[\frac{\sum_{r=1}^{R}\left[w_{r,ij}\gamma_{r,ij}^{(t-1)} + w_{r,ji}\gamma_{r,ji}^{(t-1)}\right]}{\lambda_i^{(t-1)} + \lambda_j^{(t-1)}}\right] + b}. \tag{13}$$

The derivation of the Expectation Maximization algorithm is provided in Appendix A.

## 4. Experimental Results

To evaluate the performance of our **B**ayesian **B**radley-Terry model with rater **Q**uality (**BBQ**), we conduct a series of experiments comparing it against two baselines: Bayesian Bradley-Terry (Bayes-BT) (Caron & Doucet, 2012) and a gradient descent-based BT model that incorporates rater quality (Crowd-BT) (Chen et al., 2013). For Crowd-BT, we use the implementation provided by Google Research (Google, 2025), which was employed both by CLIC (2025) and Ballé et al. (2025). The hyperparameters used in our experiments can be found in Section B.

We evaluate the performance of several Bradley–Terry (BT) variants across datasets. For natural language, we use the HUMAINE dataset (Team, 2025), a large-scale benchmark with over 100k comparisons, and MT-Bench (Zheng et al., 2023), which provides model comparison judgments from crowd workers on multi-turn dialogues. For image compression, we consider three datasets: WD (Ballé et al., 2025), a dense expert-labeled dataset with thousands of comparisons per rater; HiFiC (Mentzer et al., 2020), which evaluates learned image codecs; and ConHa (Aczel & Wattenhofer, 2024), which focuses on conditional generative

models. These datasets vary substantially in scale, number of raters, and rater expertise, providing a broad testbed for robustness. Finally, we introduce the inhomogeneous rater quality (**IHQ**) dataset[1], obtained from a user study on the CLIC2024 (CLIC, 2025) data conducted via the *Mabyduck* platform. It contains two-alternative-forced-choice (2AFC) judgments across 28 generative image compression models. To study the impact of rater quality, we provide two subsets: (i) **screened**, where raters passed routine visual prescreening checks for color vision, contrast sensitivity, and display quality, and (ii) **unscreened**, where all raters are included. The screened subset represents higher-quality raters, whereas the unscreened subset reflects the noisy conditions typical of large-scale human evaluations.

We compare each inferred ranking to a dataset-specific reference ranking, denoted by $gt$. For HUMAINE and the IHQ splits, $gt$ is given by an external reference ranking: the public HUMAINE leaderboard (ProlificAI, 2026) for HUMAINE and the official CLIC 2024 leaderboard (CLIC, 2024) for IHQ. For the remaining datasets, we approximate the ground truth by the ranking achieved on the whole dataset. We validate that this provides a reasonable approximation of ordering for real-world datasets by examining the top-1 item in each ranking. For all datasets all three models recover the same top-ranked item.

A reliable aggregation method should reproduce the same ordering if the study is repeated. We approximate this stability using bootstrapping, which provides an estimate of the variability in the rankings. This approach preserves each rater's comparison distribution and ensures a fair assessment of stability. For all datasets, we perform 10,000 bootstrap resamples of raters, except for the HUMAINE dataset, where a single bootstrap iteration for Crowd-BT takes approximately 15 minutes, as discussed in Section 4.4. On the HUMAINE dataset, we perform 1,000 resamples to reduce computation time. Note that the WD and HiFiC studies employed active selection of comparison pairs. HiFiC used a binary search strategy, while WD selected pairs based on maximum information gain. For this reason, the bootstrapping results on these two datasets should be interpreted with caution.

We evaluate performance using two metrics against $gt$. Top-1 accuracy is the fraction of bootstrapped samples that identify the same best item as $gt$. Kendall's Tau ($\tau$) measures ordinal correlation between rankings and $gt$ (Kendall, 1938), with higher values indicating better agreement. Top-1 accuracy is most relevant when the best model matters, while Kendall's Tau assesses overall agreement with the reference ranking.

---

[1]Available at `https://huggingface.co/datasets/Mabyduck/CLIC2024-test-human-eval`.

*Table 1.* Performance of different BT-based aggregation methods across several datasets. Top: Top-1 accuracy [%] compared to $gt$. Bottom: Kendall's Tau compared to $gt$. BBQ most frequently identifies the top-performing item across all datasets and achieves the strongest overall performance with respect to $gt$ on more than half of the datasets, while ranking second on the remaining datasets.

| | HUMAINE | MT-Bench | WD | HiFiC | ConHa | IHQ | | |
| | | | | | | all | scr. | unscr. |
|---|---|---|---|---|---|---|---|---|
| **Top-1 Accuracy [%]** | | | | | | | | |
| Crowd-BT | 82.70 | **100.00** | 99.29 | 98.63 | 66.59 | 85.46 | 97.24 | 32.44 |
| Bayes-BT | 97.40 | **100.00** | **100.00** | **100.00** | 57.30 | 75.30 | 98.89 | 23.59 |
| BBQ (ours) | **98.50** | **100.00** | **100.00** | **100.00** | **77.12** | **99.34** | **99.86** | **61.42** |
| **Kendall's Tau** | | | | | | | | |
| Crowd-BT | 0.8933 | **0.9743** | 0.9279 | 0.9366 | 0.9180 | 0.9210 | **0.9238** | 0.8375 |
| Bayes-BT | 0.9016 | 0.9569 | **0.9459** | 0.9293 | 0.9110 | 0.9205 | 0.9211 | 0.8371 |
| BBQ (ours) | **0.9025** | 0.9675 | 0.9359 | **0.9525** | **0.9265** | **0.9211** | 0.9204 | **0.8431** |

## 4.1. Performance Across Datasets

We compare models across datasets to assess their performance on ranking accuracy and stability. Top-1 accuracy and Kendall's Tau are summarized in Table 1. Datasets with more comparisons per model, such as MT-Bench, WD, and HiFiC, are generally easier. Interestingly, HUMAINE deviates from this trend, highlighting that factors beyond the total number of comparisons, such as rater consistency and diversity, can influence model performance.

Overall, BBQ demonstrates strong agreement and robustness across datasets. It identifies the top-performing item most frequently, being the shared best on three datasets. It recovers the overall ranking most consistently on five out of eight datasets, ranking second on the remaining three.

The three datasets where BBQ ranks second in Kendall's Tau consist of high quality, homogeneous rater sets. On MT-Bench, all models perfectly recover the top item. The WD dataset, collected by Ballé et al. (2025), includes only five raters, likely the paper's authors, suggesting exceptionally careful evaluation. In IHQ-screened, raters were explicitly filtered for quality. In such settings where rater quality is consistently high, explicitly modeling rater reliability offers little advantage, and the benefits of BBQ over simpler models are reduced.

Experiments on the IHQ dataset, considering both screened and unscreened rater subsets, reveal a clear pattern. BBQ substantially outperforms Crowd-BT and Bayes-BT when low-quality raters are present, as in IHQ-all and IHQ-unscreened. All three models achieve their best Top-1 accuracy when restricted to the screened subset, highlighting the importance of rater selection. While Crowd-BT explicitly models rater quality, its performance drops noticeably on the full dataset, likely because crowdsourced raters provide fewer than 40 comparisons each, which increases suscep-

tibility to noisy annotations. In contrast, BBQ maintains strong performance even without screening, with only a minor decrease in Top-1 accuracy, demonstrating robustness to low-quality raters.

Nonetheless, achieving uniformly high rater quality is challenging. Large-scale crowdsourcing introduces variability, screening procedures are costly, and subjective factors may affect even diligent annotators. In this context, BBQ provides a principled way to leverage partially reliable raters while reducing the impact of noisy contributions.

## 4.2. Scaling with Raters and Comparisons

As observed in Table 1, datasets with more comparisons per model generally yield better performance. The number of comparisons can be increased in two ways: by adding more raters, or by increasing the number of comparisons each rater performs.

Figure 1 illustrates the impact of both factors on the three BT variants. The left column shows performance as a function of the number of raters (with the number of comparisons per rater fixed at the maximum), while the right column shows performance as a function of the number of comparisons per rater (with the number of raters fixed at the maximum). A clear difference in performance can be observed in Top-1 accuracy, while Kendall's $\tau$ remains similar across models. Crowd-BT fails to converge with only one or two raters, highlighting the advantage of using the EM algorithm, which provides convergence guarantees compared to gradient descent.

Bayes-BT and BBQ perform similarly when the number of raters or comparisons per rater is small. BBQ requires both multiple raters and multiple comparisons per rater to effectively distinguish between rater qualities. As the number of raters or comparisons per rater increases, BBQ increas-

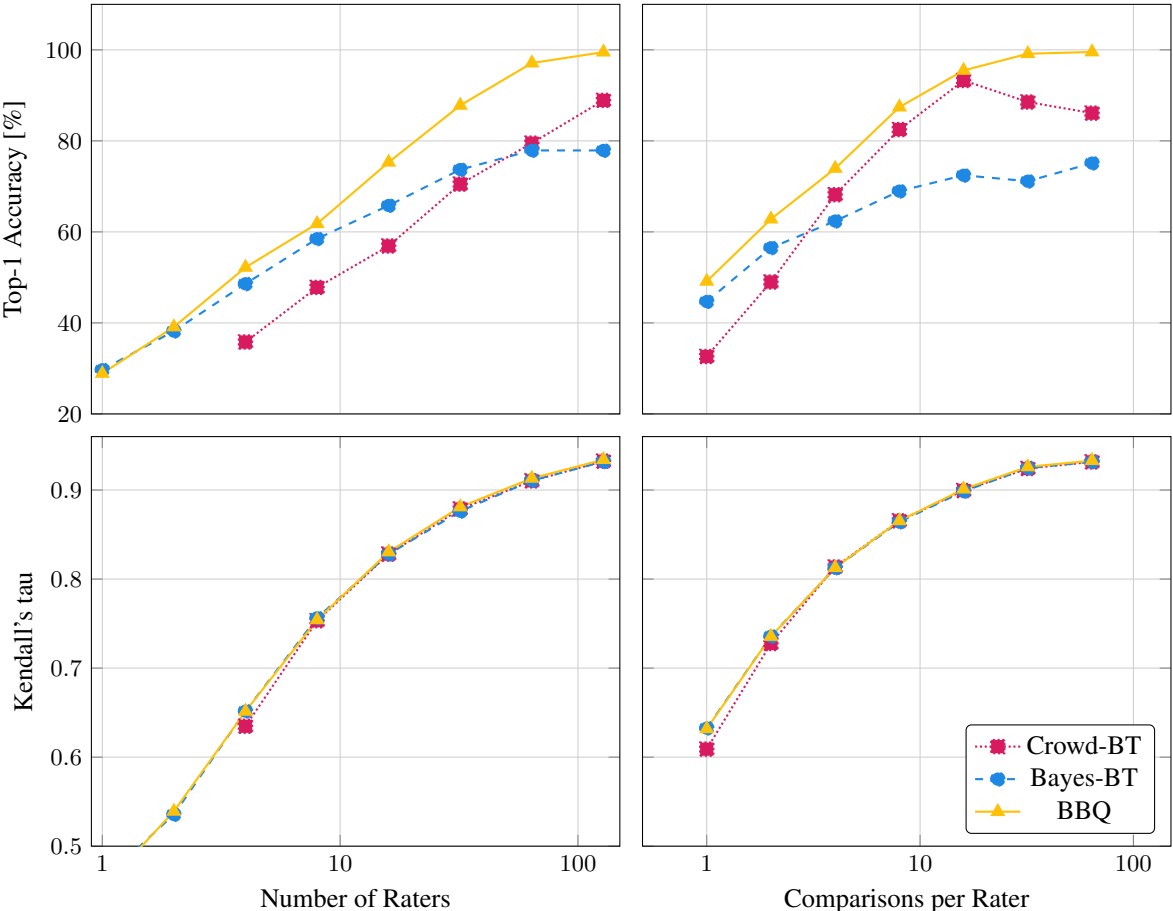

*Figure 1.* Scaling behavior of Bradley–Terry variants (Crowd-BT (Caron & Doucet, 2012), Bayes-BT (Chen et al., 2013), BBQ (ours)) on the IHQ-all dataset. **Left:** Performance vs. number of raters. **Right:** Performance vs. number of comparisons per rater. Both Top-1 accuracy and Kendall's $\tau$ improve noticeably with more raters or comparisons. While Top-1 accuracy differentiates between models, Kendall's $\tau$ remains similar across models. Crowd-BT fails to converge with very few raters, highlighting the EM algorithm's advantage. Bayes-BT and BBQ perform similarly under sparse data, but BBQ outperforms Bayes-BT as the number of raters or comparisons grows.

ingly outperforms Bayes-BT. In contrast, Bayes-BT tends to underperform overall but can surpass Crowd-BT when limited data per rater or a few number of raters are available, since Crowd-BT cannot reliably estimate rater quality in such sparse settings.

### 4.3. Rater quality

Figure 2 shows the relationship between agreement with the final ranking and predicted rater quality. We observe a clear positive correlation: raters who agree more closely with the consensus ranking are assigned higher quality by the model. For the filtered dataset, the Pearson correlation is $r = 0.551$ across 50 raters. For the unfiltered dataset, the correlation is stronger, with $r = 0.724$ across 62 raters.

As expected, the unscreened dataset contains several raters with both lower agreement and lower predicted quality. Some raters in the IHQ-unscreened dataset even fall below

the level of random guessing (50% agreement), systematically disagreeing with the majority. This highlights the importance of modeling rater quality when aggregating pairwise comparison data. BBQ successfully identifies such raters and assigns them lower quality scores, thereby reducing their influence on the final ranking and mitigating the noise they introduce.

### 4.4. Computational Efficiency

Figure 3 reports the average computation time in seconds required by each method to process a single bootstrapped sample across different datasets. These timings provide a practical perspective on the feasibility of the methods in real-world evaluation scenarios. BBQ consistently converges within a few seconds on all datasets. Crowd-BT requires more time than BBQ across the board, with particularly long runtimes on datasets with many comparisons, such as HUMAINE, where convergence takes around 15 minutes.

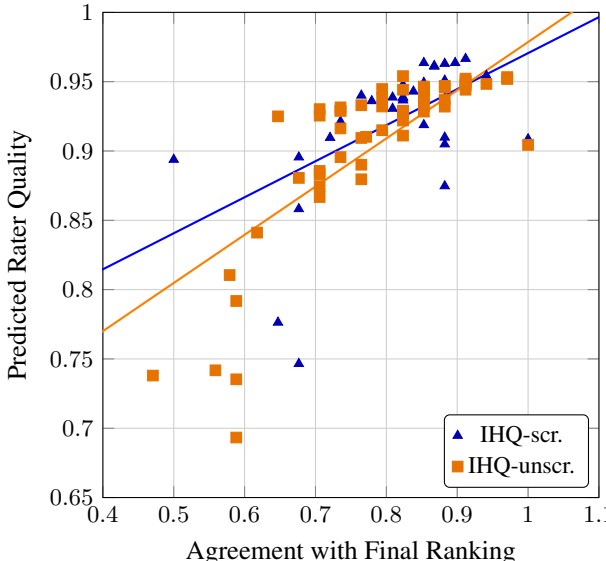

*Figure 2.* Scatter plot of rater agreement with the final ranking (x-axis) versus the predicted rater quality (y-axis) for the IHQ datasets. Each point corresponds to an individual rater. Triangles denote the filtered dataset, and squares denote the unfiltered dataset.

Bayes-BT converges slowly on the WD dataset, though on other datasets it is slightly faster than BBQ. The efficiency of BBQ stems from the closed-form EM updates derived in our method, which enable efficient inference even on large datasets.

It is also important to note that Crowd-BT was highly optimized and implemented in C (Kernighan & Ritchie, 1988), whereas BBQ was implemented using plain NumPy without specific optimizations. This suggests that BBQ could be made even faster with a compiled or vectorized implementation. Consequently, BBQ is not only more robust and stable but also highly practical for large-scale human preference studies or applications that require repeated bootstrapping. The combination of accuracy, stability, and speed makes BBQ a compelling choice for real-world deployments.

### 4.5. Confidence Intervals

We evaluate how Crowd-BT, Bayes-BT, and BBQ estimate uncertainty in item rankings. In a controlled simulation, raters provide pairwise comparisons of equally skilled items ($H_0$). Each model estimates skills and constructs uncertainty intervals to test for significant differences (Section F). The resulting Type I error rates are shown in Figure 4. At the 99% confidence level, the expected error rate is 1%. Crowd-BT and BBQ align closely, yielding slightly lower rates, while Bayes-BT is more conservative at $\sim$0.1%. Thus, Crowd-BT and BBQ provide well-calibrated uncertainty estimates, whereas Bayes-BT underestimates false positives.

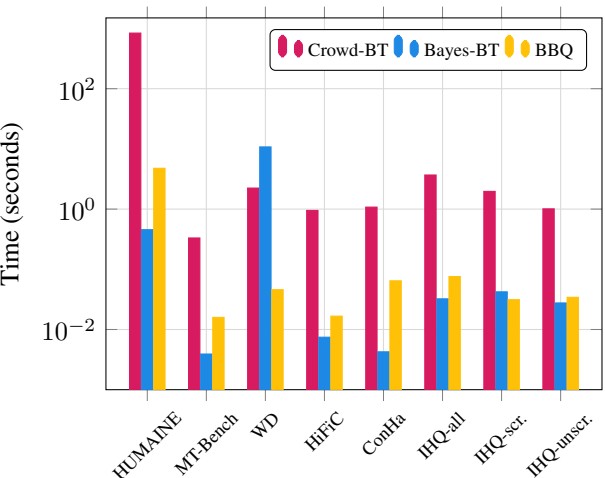

*Figure 3.* Average computation time in seconds (log-scale) for three models measured on a single bootstrapped sample across eight datasets. While Crowd-BT can require substantial computation time on some datasets, BBQ consistently remains fast across all datasets.

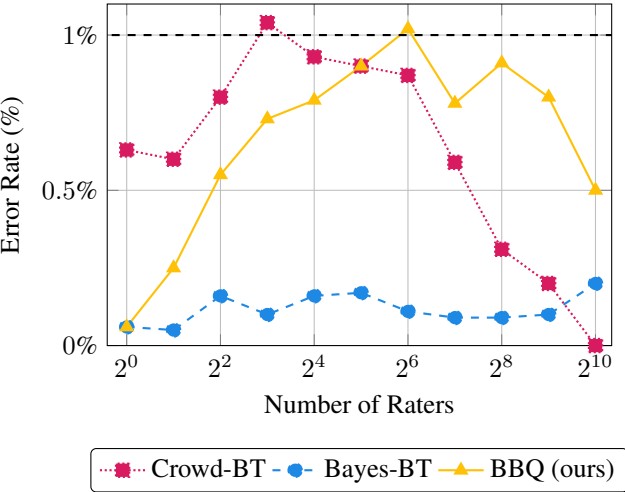

*Figure 4.* Type I error rate as a function of the number of raters. The dashed line indicates the expected error rate of 1%.

## 5. Discussion

Although BBQ effectively models rater quality, its advantages diminish in settings where all raters are uniformly reliable. In such homogeneous datasets, modeling variability provides little additional benefit. Ensuring consistently high-quality raters, however, often requires substantial cost and effort, which may not be feasible in large-scale studies. This tension highlights that BBQ is most useful in realistic crowdsourced settings, but less so when evaluations

are carefully curated. Additionally, the model assumes independence across comparisons and does not account for contextual or order effects, which may influence human judgments in practice.

BBQ also admits a straightforward extension from pairwise comparisons to rankings over multiple items. If a rater provides a ranking $i_1 \succ i_2 \succ \cdots \succ i_m$, one can apply Plackett-Luce style rank-breaking (Soufiani et al., 2014), converting the ranking into weighted implied pairwise comparisons. BBQ can then be run on these weighted comparisons with the same rater-quality parameter $q_r$, so the EM structure remains unchanged while the inputs generalize from binary comparisons to ranked lists. For applications focused only on top-$k$ or best-item identification, the same Bayesian score estimates could also be coupled with active pair selection or Thompson-style sampling (Wu & Liu, 2016).

Finally, although BBQ scales well computationally, extremely large numbers of items or raters could still pose challenges without optimized or compiled implementations.

## 6. Conclusion

We introduced BBQ, a Bayesian Bradley-Terry model that jointly estimates item quality and rater reliability. Our core contribution is the derivation of an Expectation-Maximization (EM) algorithm that simultaneously estimates item skills and individual rater quality, effectively downweighting or removing the influence of unreliable participants. By explicitly modeling rater quality, the method produces more stable and accurate rankings. The EM algorithm ensures monotonic likelihood improvement and competitive runtime, especially relative to gradient-based approaches.

Across diverse datasets, BBQ consistently achieved high Top-1 accuracy and strong Kendall's Tau, demonstrating both robustness and reliability. The model excels in scenarios with noisy or crowdsourced raters, maintaining strong performance even when raters contribute few comparisons. BBQ is particularly effective in large-scale settings where many raters are non-experts or vary widely in attentiveness and diligence. Our results highlight that incorporating rater quality is especially crucial when evaluation quality is heterogeneous or partially unreliable. Additionally, the Bayesian framework provides principled uncertainty estimates for item scores, enabling interpretable comparisons across studies. We further demonstrate that the model's error bars are well-calibrated and can be used to assess whether differences between items are statistically significant. Predicted rater quality aligns with agreement to final rankings, validating the model's ability to identify reliable evaluators.

Overall, BBQ advances human evaluation methodology by offering a practical, interpretable, and generalizable approach to aggregate noisy pairwise comparisons. This work contributes a significant step toward more cost-effective, interpretable, and reproducible human studies for evaluating AI-generated content.

## Impact Statement

This work improves the reliability of human evaluation from noisy pairwise comparisons. More robust aggregation can reduce annotation waste, quantify uncertainty, and improve benchmarks for generative models and related systems. However, more reliable preference optimization does not guarantee truthfulness, safety, or fairness, so BBQ should complement, not replace, careful study design, consent, compensation, privacy protection, and domain-specific evaluation.

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

# A. Bayesian BT with rater quality derivation

We consider $R$ raters comparing $K$ items. Each rater $r$ has quality $q_r$, meaning that with probability $q_r$ they follow the Bradley–Terry model, and with probability $1 - q_r$ they choose randomly. The following is a standard EM (Dempster et al., 1977) derivation that incorporates these rater-specific reliabilities into the Bayesian estimation framework.

## A.1. Model definition

Notation:

- $D = \{(n_{r,ij}, w_{r,ij})\}_{r=1,\dots,R; 1 \le i < j \le K}$: observed comparison counts and wins for all raters and item pairs.

- $\lambda = (\lambda_1, \dots, \lambda_K)$: skill parameters for the $K$ items.

- $q = (q_1, \dots, q_R)$: quality parameters for the $R$ raters.

- $Y_i \sim \mathcal{E}(\lambda_i)$: latent arrival time associated with item $i$.

- $A_{r,ij}^{(c)} \sim \text{Bernoulli}(q_r)$: indicator that the $c$-th comparison of pair $(i,j)$ by rater $r$ follows the Bradley-Terry model.

- $n_{r,ij}$: total number of comparisons of $(i,j)$ by rater $r$.

- $n_{ij} = \sum_{r=1}^{R} n_{r,ij}$: total number of comparisons of $(i,j)$.

- $n_r = \sum_{i=1}^{K} \sum_{j=i+1}^{K} n_{r,ij}$: total number of comparisons made by rater $r$.

- $w_{r,ij}$: number of times rater $r$ ranked $i$ above $j$.

- $w_{ij} = \sum_{r=1}^{R} w_{r,ij}$: total number of times $i$ was ranked above $j$.

- $m_{r,ij} = \sum_{c=1}^{n_{r,ij}} A_{r,ij}^{(c)}$: number of Bradley-Terry-model comparisons of $(i,j)$ by rater $r$.

- $m_{ij} = \sum_{r=1}^{R} m_{r,ij}$: total number of Bradley-Terry-model comparisons of $(i,j)$.

- $v_{r,ij} = \sum_{c=1}^{w_{r,ij}} A_{r,ij}^{(c)}$: number of Bradley-Terry-model wins of $i$ over $j$ by rater $r$.

- $v_{ir} = \sum_{j \neq i} v_{r,ij}$: total number of Bradley-Terry-model wins of item $i$ attributed to rater $r$.

- $v_{ij} = \sum_{r=1}^{R} v_{r,ij}$: total number of Bradley-Terry-model wins of $i$ over $j$.

- $Z_{r,ij} = \sum_{c=1}^{n_{r,ij}} A_{r,ij}^{(c)} \min(Y_i^{(c)}, Y_j^{(c)})$: sum of minimal arrival times, with

$$Z_{r,ij} \mid m_{r,ij} \sim \Gamma(m_{r,ij}, \lambda_i + \lambda_j).$$

- $Z_{ij} = \sum_{r=1}^{R} Z_{r,ij}$ and $z_{ij}$ its realization: total minimal arrival time across raters for pair $(i,j)$.

Bradley-Terry probability:

$$P(i \text{ beats } j) = \frac{\lambda_i}{\lambda_i + \lambda_j}.$$

Mixture with rater quality:

$$P(r \text{ ranks } i \text{ above } j) = q_r \frac{\lambda_i}{\lambda_i + \lambda_j} + (1 - q_r)\frac{1}{2}.$$

Latent exponential view:

$$Y_i \sim \mathcal{E}(\lambda_i), \quad Y_j \sim \mathcal{E}(\lambda_j), \quad P(i \text{ beats } j) = P(Y_i < Y_j).$$

### A.2. Complete-Data Log-Likelihood

We need to compute:

$$\ell_c(\lambda, q; D, Z, A) = \log P(D, Z, A \mid \lambda, q)$$
$$= \log P(Z \mid D, A, \lambda, q) + \log P(D \mid A, \lambda, q) + \log P(A \mid \lambda, q).$$

By conditional independence, $Z \perp q \mid (D, A, \lambda)$, $D \perp q \mid (A, \lambda)$, and $A \perp (D, \lambda) \mid q$, so the three factors simplify to $P(Z \mid A, \lambda)$, $P(D \mid A, \lambda)$, and $P(A \mid q)$, respectively.

Log-Likelihood of $P(Z \mid A, \lambda)$:

$$\log P(Z \mid A, \lambda) = \sum_{i=1}^{K} \sum_{j=i+1}^{K} \left[ m_{ij} \log(\lambda_i + \lambda_j) - (\lambda_i + \lambda_j) z_{ij} + (m_{ij} - 1) \log z_{ij} - \log \Gamma(m_{ij}) \right]$$

Log-Likelihood of $P(D \mid A, \lambda)$:

$$\log P(D \mid A, \lambda) = \sum_{r=1}^{R} \sum_{i=1}^{K} \sum_{\substack{j=1 \\ j \neq i}}^{K} \left[ v_{r,ij} \log \frac{\lambda_i}{\lambda_i + \lambda_j} + (w_{r,ij} - v_{r,ij}) \log \frac{1}{2} \right]$$

$$= \sum_{r=1}^{R} \sum_{i=1}^{K} [v_{ir} \log \lambda_i] - \sum_{r=1}^{R} \sum_{i=1}^{K} \sum_{j=i+1}^{K} \left[ (v_{r,ij} + v_{r,ji}) \log (\lambda_i + \lambda_j) + (w_{r,ij} + w_{r,ji} - v_{r,ij} - v_{r,ji}) \log 2 \right]$$

$$= \sum_{i=1}^{K} [v_i \log \lambda_i] - \sum_{i=1}^{K} \sum_{j=i+1}^{K} [m_{ij} \log (\lambda_i + \lambda_j) + (n_{ij} - m_{ij}) \log 2]$$

Log-Likelihood of $P(A \mid q)$:

$$\log P(A \mid q) = \sum_{r=1}^{R} \sum_{i=1}^{K} \sum_{j=i+1}^{K} \left[ m_{r,ij} \log q_r + (n_{r,ij} - m_{r,ij}) \log(1 - q_r) \right]$$

Complete-Data Log-Likelihood:

$$\ell_c(\lambda, q; D, Z, A) = \log P(Z \mid A, \lambda) + \log P(D \mid A, \lambda) + \log P(A \mid q)$$

$$= \sum_{i=1}^{K} \sum_{j=i+1}^{K} \left[ m_{ij} \log(\lambda_i + \lambda_j) - (\lambda_i + \lambda_j) z_{ij} + (m_{ij} - 1) \log z_{ij} - \log \Gamma(m_{ij}) \right]$$

$$+ \sum_{i=1}^{K} [v_i \log \lambda_i] - \sum_{i=1}^{K} \sum_{j=i+1}^{K} [m_{ij} \log (\lambda_i + \lambda_j) + (n_{ij} - m_{ij}) \log 2]$$

$$+ \sum_{r=1}^{R} \sum_{i=1}^{K} \sum_{j=i+1}^{K} \left[ m_{r,ij} \log q_r + (n_{r,ij} - m_{r,ij}) \log(1 - q_r) \right]$$

$$= \sum_{i=1}^{K} \sum_{j=i+1}^{K} \left[ - (n_{ij} - m_{ij}) \log 2 - (\lambda_i + \lambda_j) z_{ij} + (m_{ij} - 1) \log z_{ij} - \log \Gamma(m_{ij}) \right]$$

$$+ \sum_{r=1}^{R} \sum_{i=1}^{K} \sum_{j=i+1}^{K} \left[ m_{r,ij} \log q_r + (n_{r,ij} - m_{r,ij}) \log(1 - q_r) \right] + \sum_{i=1}^{K} [v_i \log \lambda_i]$$

### A.3. Expectation step

We introduce conjugate priors: Gamma distribution $\lambda_i \sim \Gamma(a, b)$ for each item $i$, and Beta $q_r \sim B(\alpha, \beta)$ for each rater $r$.

The $Q$-function is the expectation of the complete-data log-posterior:

$$Q(\lambda, q \mid \lambda^*, q^*) = \mathbb{E}_{A,Z|D,\lambda^*,q^*}\left[\ell_c(\lambda, q; D, Z, A) + \log P(\lambda) + \log P(q)\right].$$

Posterior probability of a Bradley-Terry-consistent annotation:

$$P\big(A_{r,ij}^{(k)} = 1 \mid i \succ j, D, \lambda^*, q^*\big) = \frac{P(i \succ j \mid A = 1)\, P(A = 1)}{P(i \succ j \mid A = 1)\, P(A = 1) \ + \ P(i \succ j \mid A = 0)\, P(A = 0)}$$

$$= \frac{\left(\frac{\lambda_i^*}{\lambda_i^* + \lambda_j^*}\right) q_r^*}{\left(\frac{\lambda_i^*}{\lambda_i^* + \lambda_j^*}\right) q_r^* \ + \ \frac{1}{2}\left(1 - q_r^*\right)} \ = \ \gamma_{r,ij}^*.$$

Expected sufficient statistics:

$$\mathbb{E}[m_{r,ij} \mid D, \lambda^*, q^*] = w_{r,ij}\gamma_{r,ij}^* + w_{r,ji}\gamma_{r,ji}^*$$

$$\mathbb{E}[m_{ij} \mid D, \lambda^*, q^*] = \sum_{r=1}^{R}(w_{r,ij}\gamma_{r,ij}^* + w_{r,ji}\gamma_{r,ji}^*)$$

$$\mathbb{E}[v_{r,ij} \mid D, \lambda^*, q^*] = w_{r,ij}\,\gamma_{r,ij}^*$$

$$\mathbb{E}[v_{ir} \mid D, \lambda^*, q^*] = \sum_{\substack{j=1 \\ j \neq i}}^{K} w_{r,ij}\,\gamma_{r,ij}^*$$

$$\mathbb{E}[v_i \mid D, \lambda^*, q^*] = \sum_{r=1}^{R}\sum_{\substack{j=1 \\ j \neq i}}^{K} w_{r,ij}\,\gamma_{r,ij}^*$$

$$\mathbb{E}[z_{ij} \mid D, \lambda^*, q^*] = \frac{\sum_{r=1}^{R}(w_{r,ij}\gamma_{r,ij}^* + w_{r,ji}\gamma_{r,ji}^*)}{\lambda_i^* + \lambda_j^*}.$$

Expected complete-data log-posterior:

$$\mathbb{E}[\ell_c] = \sum_{i=1}^{K}\sum_{j=i+1}^{K}\left[-(\lambda_i + \lambda_j)\frac{\sum_{r=1}^{R}(w_{r,ij}\gamma_{r,ij}^* + w_{r,ji}\gamma_{r,ji}^*)}{\lambda_i^* + \lambda_j^*}\right]$$

$$+ \sum_{r=1}^{R}\sum_{i=1}^{K}\sum_{j=i+1}^{K}\left[(w_{r,ij}\gamma_{r,ij}^* + w_{r,ji}\gamma_{r,ji}^*)\log q_r + (n_{r,ij} - (w_{r,ij}\gamma_{r,ij}^* + w_{r,ji}\gamma_{r,ji}^*))\log(1 - q_r)\right]$$

$$+ \sum_{r=1}^{R}\sum_{i=1}^{K}\sum_{\substack{j=1 \\ j \neq i}}^{K} w_{r,ij}\gamma_{r,ij}^* \log \lambda_i + \text{const}.$$

Priors contribute:

$$\mathbb{E}[\log P(\lambda)] = \sum_{i=1}^{K}\left[(a - 1)\log \lambda_i - b\lambda_i\right],$$

$$\mathbb{E}[\log P(q)] = \sum_{r=1}^{R}\left[(\alpha - 1)\log q_r + (\beta - 1)\log(1 - q_r)\right].$$

Final $Q$-function:

$$Q(\lambda, q \mid \lambda^*, q^*) = \sum_{i=1}^{K} \sum_{j=i+1}^{K} \left[ -(\lambda_i + \lambda_j) \frac{\sum_{r=1}^{R}(w_{r,ij}\gamma_{r,ij}^* + w_{r,ji}\gamma_{r,ji}^*)}{\lambda_i^* + \lambda_j^*} \right]$$

$$+ \sum_{r=1}^{R} \sum_{i=1}^{K} \sum_{j=i+1}^{K} \left[ (w_{r,ij}\gamma_{r,ij}^* + w_{r,ji}\gamma_{r,ji}^*) \log q_r + (n_{r,ij} - (w_{r,ij}\gamma_{r,ij}^* + w_{r,ji}\gamma_{r,ji}^*)) \log(1 - q_r) \right]$$

$$+ \sum_{r=1}^{R} \sum_{i=1}^{K} \sum_{\substack{j=1 \\ j \neq i}}^{K} w_{r,ij}\gamma_{r,ij}^* \log \lambda_i + \sum_{i=1}^{K} [(a-1)\log\lambda_i - b\lambda_i] + \sum_{r=1}^{R} [(\alpha-1)\log q_r + (\beta-1)\log(1-q_r)]$$

$$+ \text{const.}$$

where

$$\gamma_{r,ij}^* = \frac{q_r^* \dfrac{\lambda_i^*}{\lambda_i^* + \lambda_j^*}}{q_r^* \dfrac{\lambda_i^*}{\lambda_i^* + \lambda_j^*} + (1 - q_r^*)\frac{1}{2}}.$$

### A.4. M-step

The M-step maximizes the $Q$-function w.r.t. the parameters $(\lambda, q)$, holding the expectations computed in the E-step fixed.

**Update for $q_r$**  The update for each rater quality $q_r$ is obtained by maximizing $Q$ with respect to $q_r$ (including the Beta prior).

$$Q(q \mid \lambda^*, q^*) = \sum_{r=1}^{R} \sum_{i=1}^{K} \sum_{j=i+1}^{K} \left[ (w_{r,ij}\gamma_{r,ij}^* + w_{r,ji}\gamma_{r,ji}^*) \log q_r + (n_{r,ij} - (w_{r,ij}\gamma_{r,ij}^* + w_{r,ji}\gamma_{r,ji}^*)) \log(1 - q_r) \right]$$

$$+ (\alpha - 1)\log q_r + (\beta - 1)\log(1 - q_r) + \text{const.}$$

$$\frac{\partial Q}{\partial q_r} = \sum_{i=1}^{K} \sum_{j=i+1}^{K} \left[ \frac{w_{r,ij}\gamma_{r,ij}^* + w_{r,ji}\gamma_{r,ji}^*}{q_r} - \frac{n_{r,ij} - (w_{r,ij}\gamma_{r,ij}^* + w_{r,ji}\gamma_{r,ji}^*)}{1 - q_r} \right] + \frac{\alpha - 1}{q_r} - \frac{\beta - 1}{1 - q_r}$$

$$0 \stackrel{!}{=} \frac{\sum_{i=1}^{K} \sum_{j=i+1}^{K}(w_{r,ij}\gamma_{r,ij}^* + w_{r,ji}\gamma_{r,ji}^*) + (\alpha - 1)}{q_r}$$

$$- \frac{\sum_{i=1}^{K} \sum_{j=i+1}^{K}(n_{r,ij} - (w_{r,ij}\gamma_{r,ij}^* + w_{r,ji}\gamma_{r,ji}^*)) + (\beta - 1)}{1 - q_r}$$

$$\implies (1 - q_r)\left( \sum_{i=1}^{K} \sum_{j=i+1}^{K}(w_{r,ij}\gamma_{r,ij}^* + w_{r,ji}\gamma_{r,ji}^*) + (\alpha - 1) \right)$$

$$\implies q_r\left( \sum_{i=1}^{K} \sum_{j=i+1}^{K}(n_{r,ij} - (w_{r,ij}\gamma_{r,ij}^* + w_{r,ji}\gamma_{r,ji}^*)) + (\beta - 1) \right)$$

$$\implies q_r = \frac{\sum_{i=1}^{K} \sum_{j=i+1}^{K}(w_{r,ij}\gamma_{r,ij}^* + w_{r,ji}\gamma_{r,ji}^*) + (\alpha - 1)}{\sum_{i=1}^{K} \sum_{j=i+1}^{K}(w_{r,ij}\gamma_{r,ij}^* + w_{r,ji}\gamma_{r,ji}^* + n_{r,ij} - (w_{r,ij}\gamma_{r,ij}^* + w_{r,ji}\gamma_{r,ji}^*)) + (\beta + \alpha - 2)}$$

$$\implies q_r^{(t)} = \frac{\sum_{i=1}^{K} \sum_{j=i+1}^{K}(w_{r,ij}\gamma_{r,ij}^{(t-1)} + w_{r,ji}\gamma_{r,ji}^{(t-1)}) + (\alpha - 1)}{n_r + \beta + \alpha - 2}$$

where the last line gives the explicit update at iteration $t$.

**Update for $\lambda_i$** The update for each item skill $\lambda_i$ is obtained by maximizing $Q$ with respect to $\lambda_i$ (including the Gamma prior).

$$Q(\lambda \mid \lambda^*, q^*) = \sum_{i=1}^{K} \sum_{j=i+1}^{K} -(\lambda_i + \lambda_j) \frac{\sum_{r=1}^{R}(w_{r,ij}\gamma_{r,ij}^* + w_{r,ji}\gamma_{r,ji}^*)}{\lambda_i^* + \lambda_j^*}$$
$$+ \sum_{r=1}^{R}\sum_{i=1}^{K}\sum_{\substack{j=1 \\ j\neq i}}^{K} w_{r,ij}\gamma_{r,ij}^* \log \lambda_i$$
$$+ \sum_{i=1}^{K} \left[(a-1)\log\lambda_i - b\lambda_i\right] + \text{const.}$$

$$\frac{\partial Q}{\partial \lambda_i} = -\sum_{\substack{j=1 \\ j\neq i}}^{K} \frac{\sum_{r=1}^{R}(w_{r,ij}\gamma_{r,ij}^* + w_{r,ji}\gamma_{r,ji}^*)}{\lambda_i^* + \lambda_j^*} + \sum_{r=1}^{R} \frac{\sum_{\substack{j=1 \\ j\neq i}}^{K} w_{r,ij}\gamma_{r,ij}^*}{\lambda_i} + \frac{a-1}{\lambda_i} - b$$

$$0 \overset{!}{=} \frac{\sum_{r=1}^{R}\sum_{\substack{j=1 \\ j\neq i}}^{K} w_{r,ij}\gamma_{r,ij}^* + (a-1)}{\lambda_i} - \sum_{\substack{j=1 \\ j\neq i}}^{K} \frac{\sum_{r=1}^{R}(w_{r,ij}\gamma_{r,ij}^* + w_{r,ji}\gamma_{r,ji}^*)}{\lambda_i^* + \lambda_j^*} - b$$

$$\implies \quad \lambda_i^{(t)} = \frac{\sum_{r=1}^{R}\sum_{\substack{j=1 \\ j\neq i}}^{K} w_{r,ij}\gamma_{r,ij}^{(t-1)} + (a-1)}{\sum_{\substack{j=1 \\ j\neq i}}^{K} \frac{\sum_{r=1}^{R}(w_{r,ij}\gamma_{r,ij}^{(t-1)} + w_{r,ji}\gamma_{r,ji}^{(t-1)})}{\lambda_i^{(t-1)} + \lambda_j^{(t-1)}} + b}$$

$$\gamma_{r,ij}^{(t-1)} = \frac{q_r^{(t-1)} \frac{\lambda_i^{(t-1)}}{\lambda_i^{(t-1)} + \lambda_j^{(t-1)}}}{q_r^{(t-1)} \frac{\lambda_i^{(t-1)}}{\lambda_i^{(t-1)} + \lambda_j^{(t-1)}} + (1 - q_r^{(t-1)})\frac{1}{2}}$$

where the last line gives the explicit update at iteration $t$.

## B. Hyperparameters

For Crowd-BT, we use the implementation of Google (2025) with the default hyperparameters.

The only hyperparameters in the Bayes-BT and BBQ models are the prior distribution parameters and the stopping thresholds. For both Bayes-BT and BBQ, we consider the model converged when no ELO score changes by more than 1 between two iterations. The ELO score can be calculated from the skill parameter $\lambda$ as:

$$\text{ELO} = \log(\text{skill}) \cdot \text{ELO\_SCALE\_FACTOR}. \tag{14}$$

where we set the ELO_SCALE_FACTOR to 400.

We chose a gamma prior with shape 5 and rate 0.1 for the skill parameters in both Bayes-BT and BBQ. For BBQ, the beta prior on the rater quality has $\alpha = 10$ and $\beta = 2$. These settings are used for all main experiments. The ablations in Tables 2 to 5 show that the main conclusions are stable across a broad range of conjugate prior hyperparameters.

## C. Prior Sensitivity

We ablate the conjugate prior hyperparameters used by Bayes-BT and BBQ on IHQ-all and ConHa. For the Gamma prior tables, rows vary the shape $a$ and columns vary the rate $b$. For the Beta prior tables, rows vary $\alpha$ and columns vary $\beta$, while

*Table 2.* Prior sensitivity on IHQ-all: Top-1 accuracy [%].

| Bayes-BT, Gamma prior | | | | | BBQ, Gamma prior | | | | | BBQ, Beta prior | | |
|---|---|---|---|---|---|---|---|---|---|---|---|---|
| $a \backslash b$ | 0.02 | 0.1 | 0.5 | 2 | $a \backslash b$ | 0.02 | 0.1 | 0.5 | 2 | $\alpha \backslash \beta$ | 2 | 10 | 50 |
| 5 | 76.30 | 76.90 | 73.50 | 77.20 | 5 | 99.40 | 99.10 | 99.70 | 98.90 | 2 | 100.00 | 100.00 | 100.00 |
| 20 | 91.10 | 92.60 | 92.00 | 92.10 | 20 | 99.00 | 99.80 | 99.40 | 99.30 | 10 | 99.00 | 99.80 | 100.00 |
| 100 | 99.90 | 99.50 | 100.00 | 99.90 | 100 | 100.00 | 100.00 | 100.00 | 99.80 | 50 | 90.00 | 99.20 | 100.00 |
| | | | | | | | | | | 200 | 80.10 | 92.20 | 99.50 |

*Table 3.* Prior sensitivity on IHQ-all: Kendall's $\tau$.

| Bayes-BT, Gamma prior | | | | | BBQ, Gamma prior | | | | | BBQ, Beta prior | | |
|---|---|---|---|---|---|---|---|---|---|---|---|---|
| $a \backslash b$ | 0.02 | 0.1 | 0.5 | 2 | $a \backslash b$ | 0.02 | 0.1 | 0.5 | 2 | $\alpha \backslash \beta$ | 2 | 10 | 50 |
| 5 | 0.9237 | 0.9239 | 0.9242 | 0.9237 | 5 | 0.9268 | 0.9275 | 0.9270 | 0.9266 | 2 | 0.9281 | 0.9216 | 0.8914 |
| 20 | 0.9223 | 0.9221 | 0.9228 | 0.9219 | 20 | 0.9223 | 0.9230 | 0.9218 | 0.9225 | 10 | 0.9265 | 0.9268 | 0.9171 |
| 100 | 0.9064 | 0.9043 | 0.9050 | 0.9056 | 100 | 0.9052 | 0.9069 | 0.9043 | 0.9054 | 50 | 0.9243 | 0.9260 | 0.9237 |
| | | | | | | | | | | 200 | 0.9246 | 0.9245 | 0.9245 |

the Gamma prior is fixed at $a = 5$ and $b = 0.1$.

## D. Datasets

We evaluate our models on a diverse set of human preference datasets covering both language and image domains. Table 6 provides an overview.

## E. User Study Platform

All pairwise comparisons on the CLIC2024 (CLIC, 2025) dataset were collected using the Mabyduck platform (Ltd., 2025). A screenshot of the platform can be seen in Figure 5. The task asks: "Which image looks more similar to the reference image?" Ties are not allowed, and all pairs were selected uniformly at random.

To study rater quality, we split the IHQ dataset into screened and unscreened subsets. Screened raters passed routine visual pre-tests for color vision, display contrast, and sensitivity to subtle differences, and completed training comparisons choosing the higher-quality image to reinforce evaluation criteria. Sessions were short (typically under about 10 minutes), so within-session attention drift is likely limited.

Figure 6 shows four example images from the pre-screening process. The first two are standard color blindness tests, where raters must correctly identify the number shown in each pattern. The final two images are shape detection tests designed to evaluate the raters' sensitivity to low-contrast objects: one features a light gray shape on a white background, and the other a dark gray shape on a black background. These pre-screening tests help ensure that only raters with adequate visual capabilities contribute to the screened subset.

## F. Uncertainty Estimation Details

In addition to ranking items, estimating uncertainty is important to assess whether observed differences are statistically significant.

The Bayesian Bradley-Terry (Bayes-BT) and Bayesian Bradley-Terry with Quality (BBQ) models quantify uncertainty via the posterior distribution over item skills. Each skill has a Gamma prior, which is updated using the observed pairwise comparisons. Credible intervals derived from the posterior are then converted to the Elo scale to facilitate comparison across items.

The classical Crowd-BT model, being non-Bayesian, estimates uncertainty using a frequentist approximation. Specifically, a second-order Taylor expansion around the maximum likelihood estimate is employed. The Hessian of the log-likelihood is inverted to obtain the covariance matrix, whose diagonal entries correspond to the variances of individual items. These variances are converted to 99% confidence intervals via:

$$p_{99} = \sqrt{\text{diag(covariance)}} \times k_{\text{Erfc0\_01}} \times \sqrt{2} \approx \sqrt{\text{diag(covariance)}} \times 3.29,$$

*Table 4.* Prior sensitivity on ConHa: Top-1 accuracy [%].

| Bayes-BT, Gamma prior | | | | | BBQ, Gamma prior | | | | | BBQ, Beta prior | | | |
|---|---|---|---|---|---|---|---|---|---|---|---|---|---|
| $a \backslash b$ | 0.02 | 0.1 | 0.5 | 2 | $a \backslash b$ | 0.02 | 0.1 | 0.5 | 2 | $\alpha \backslash \beta$ | 2 | 10 | 50 |
| 5 | 56.30 | 56.50 | 57.70 | 55.40 | 5 | 77.90 | 77.50 | 79.30 | 78.30 | 2 | 71.20 | 77.80 | 80.20 |
| 20 | 65.10 | 62.00 | 63.20 | 62.00 | 20 | 76.50 | 78.50 | 77.80 | 77.00 | 10 | 78.80 | 80.30 | 85.90 |
| 100 | 72.80 | 72.00 | 70.40 | 72.80 | 100 | 76.40 | 77.40 | 73.20 | 73.90 | 50 | 64.50 | 75.30 | 82.70 |
| | | | | | | | | | | 200 | 58.60 | 62.00 | 76.00 |

*Table 5.* Prior sensitivity on ConHa: Kendall's $\tau$.

| Bayes-BT, Gamma prior | | | | | BBQ, Gamma prior | | | | | BBQ, Beta prior | | | |
|---|---|---|---|---|---|---|---|---|---|---|---|---|---|
| $a \backslash b$ | 0.02 | 0.1 | 0.5 | 2 | $a \backslash b$ | 0.02 | 0.1 | 0.5 | 2 | $\alpha \backslash \beta$ | 2 | 10 | 50 |
| 5 | 0.9072 | 0.9091 | 0.9109 | 0.9082 | 5 | 0.9256 | 0.9244 | 0.9293 | 0.9271 | 2 | 0.9212 | 0.9238 | 0.8951 |
| 20 | 0.9101 | 0.9106 | 0.9084 | 0.9081 | 20 | 0.9253 | 0.9265 | 0.9267 | 0.9257 | 10 | 0.9289 | 0.9245 | 0.9301 |
| 100 | 0.9067 | 0.9046 | 0.9000 | 0.9034 | 100 | 0.9036 | 0.9111 | 0.9053 | 0.9078 | 50 | 0.9178 | 0.9203 | 0.9279 |
| | | | | | | | | | | 200 | 0.9144 | 0.9142 | 0.9171 |

where the constants scale the standard deviation to match the 99% confidence level. Narrower likelihood peaks yield smaller intervals, reflecting higher certainty, while flatter peaks produce larger intervals, indicating greater uncertainty.

To evaluate how well these models estimate uncertainty, we conduct a controlled simulation experiment. We generate trials with two items of equal skill (50–50 win probability) and simulate pairwise comparison data using coin flips. For each trial, multiple raters (users) are simulated, and models are applied to estimate Elo scores and their associated uncertainty. This two-item, equal-skill design is intentional: it isolates Type I error calibration under the null hypothesis rather than attempting to mimic the full heterogeneity of real evaluation datasets.

Formally, the null hypothesis $H_0$ states that the two items are equally strong. For each trial, we compute the 99% confidence (or credible) interval for each item's skill estimate. A model is said to incorrectly reject $H_0$ if the confidence intervals do not overlap, indicating a statistically significant difference between items when none exists. The primary metric of interest is the frequency with which each model incorrectly concludes that the items are different, i.e., the observed Type I error rate.

We perform 10,000 trials, using 50 comparisons per rater, while varying the number of raters to assess how uncertainty estimates scale with the amount of data. This setup allows us to estimate the empirical Type I error rate for each model and compare it against the theoretical expectation of 1% at the 99% confidence level. As shown in Figure 4, Crowd-BT and BBQ exhibit Type I error rates close to the expected 1%, indicating that their uncertainty estimates are well-calibrated. In contrast, Bayes-BT is overly conservative, consistently producing lower error rates than expected, which suggests its credible intervals are not as well-calibrated for significance testing.

*Table 6.* Summary of datasets used in our experiments. The IHQ dataset was collected by us on the Mabyduck platform and includes both 2AFC and 3AFC settings. For this dataset, we provide screened and unscreened subsets: screened subsets include only raters who passed routine visual pre-screening tests for color vision, contrast sensitivity, and display quality, ensuring higher reliability, while unscreened subsets include all raters.

| dataset | # comparisons | # raters | # items | comparisons/rater | comparisons/item |
|---|---|---|---|---|---|
| HUMAINE (Team, 2025) | 105,220 | 1,977 | 27 | 53.2 | 3897.0 |
| MT-Bench (Zheng et al., 2023) | 3,355 | 65 | 6 | 51.6 | 559.2 |
| WD (Ballé et al., 2025) | 16,659 | 5 | 30 | 3331.8 | 555.3 |
| HiFiC (Mentzer et al., 2020) | 5,220 | 20 | 9 | 261.0 | 580.0 |
| ConHa (Aczel & Wattenhofer, 2024) | 1,531 | 40 | 8 | 38.3 | 191.4 |
| IHQ-all | 4,074 | 112 | 28 | 36.4 | 145.5 |
| IHQ-screened | 2,012 | 50 | 28 | 40.2 | 71.9 |
| IHQ-unscreened | 2,062 | 62 | 28 | 33.3 | 73.6 |

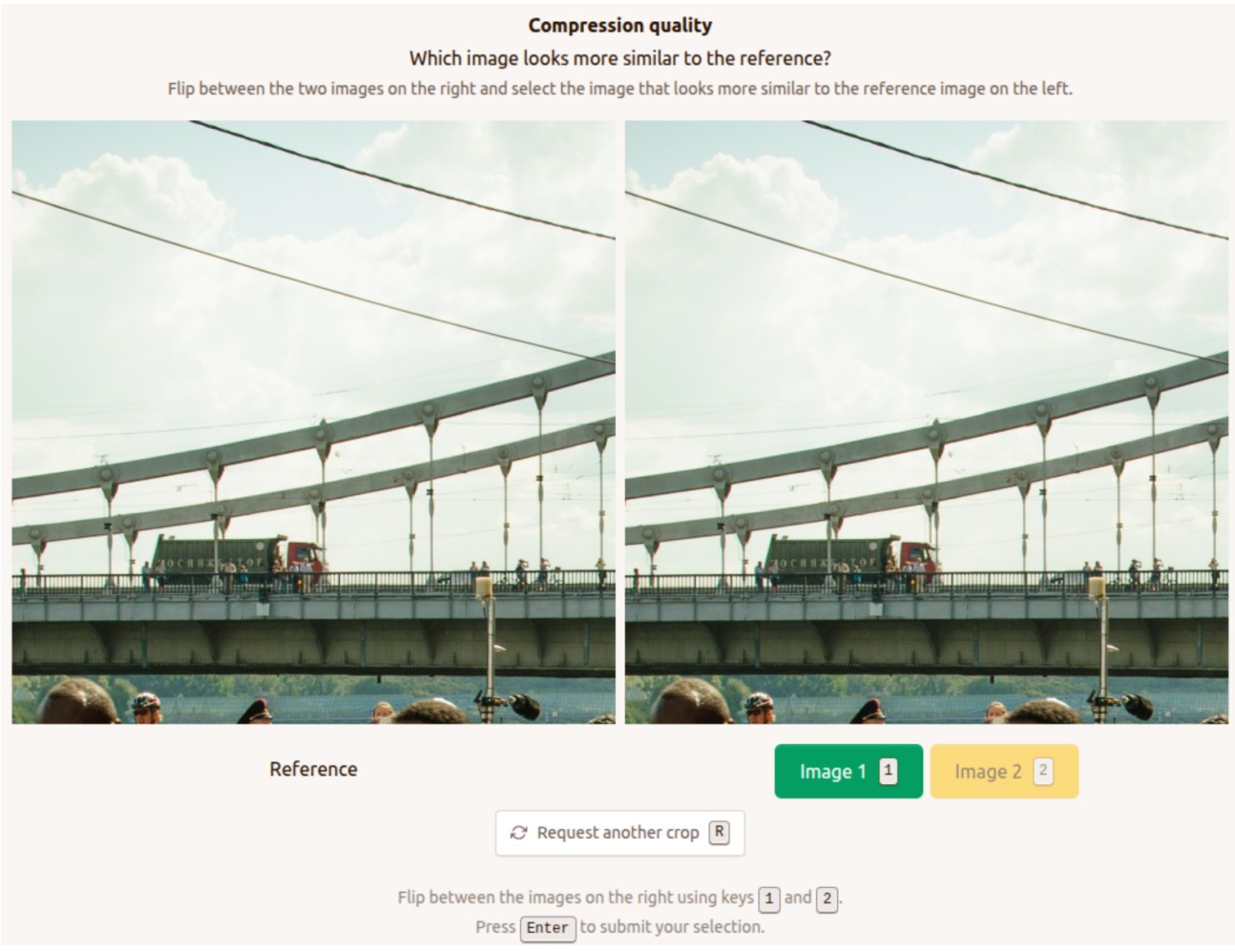

*Figure 5.* Screenshot of the Mabyduck user study platform used for collecting pairwise comparisons. A reference image is shown on the left, and the rater selects between two compressed images on the right.

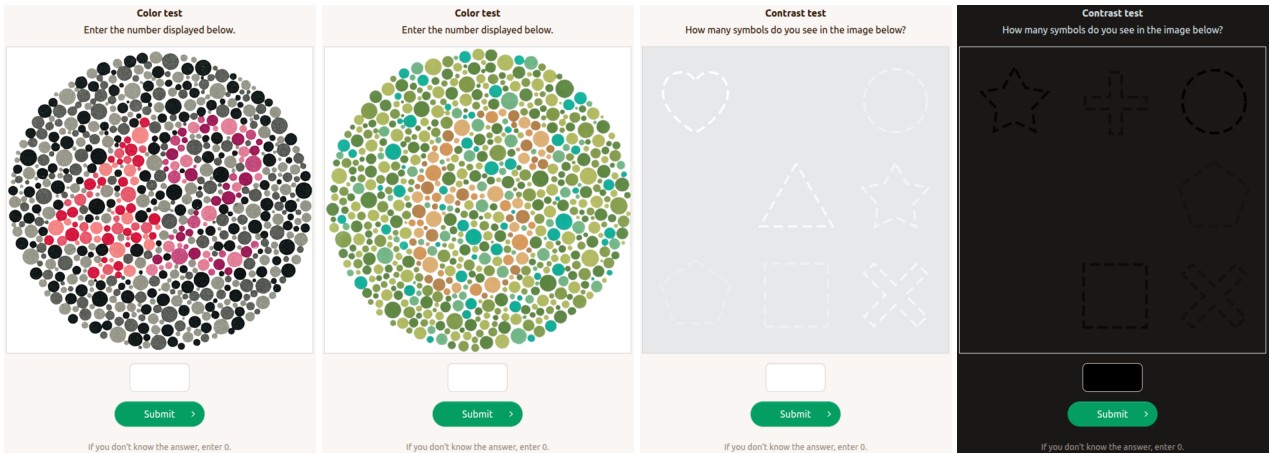

*Figure 6.* Four example images from the pre-screening process for raters. The first two are color blindness tests, where raters must identify the number displayed in each pattern. The last two are shape detection tests designed to evaluate sensitivity to low-contrast objects: one light gray shape on a white background and one dark gray shape on a black background.

