# OpenReview forum: "Efficient Bayesian Inference from Noisy Pairwise Comparisons"
_ICML.cc/2026/Conference — ICML 2026 regular_

### Official Review · Reviewer_vjFj · 2026-02-17

**Soundness:** 1
**Presentation:** 3
**Significance:** 2
**Originality:** 2
**Overall Recommendation:** 3
**Confidence:** 3

**Summary:**

The authors study pairwise comparisons (and the corresponding full rankings) in the setting where there are multiple raters and the reliability of the raters vary. The authors propose a Bayesian Bradley-Terry model that incorporate rater quality (BBQ). The authors show theoretical convergence guarantees associated with the proposed model, and applies the model on a range of datasets, comparing it with the related Bayes-BT and Crowd-BT models. The results show that the rankings from the BBQ model are the most self consistent under bootstrapping the data samples.

**Compliance With Llm Reviewing Policy:**

Affirmed.

**Final Justification:**

The rebuttal partly addressed my concerns in the paper. While I still think the evaluation protocol is rather weak, the EM formulation is solid and I have hence decided to raise my score to 3.

**Key Questions For Authors:**

1. Please address the major weakness 1. in detail.
2. This is more of a suggestion: the evaluation protocol would be much stronger if you had access to ground truth. For instance by viewing different LLMs as raters, and using a dataset with known ground truth, you could see how well aligned the BBQ model is with the true accuracy ratings.
3. Can the analysis be extended to the setting where each rater ranks N items instead of pairwise?

**Limitations:**

yes

**Strengths And Weaknesses:**

Strengths:
1. The EM formulation appears to be solid.
2. The writing in the paper is mostly clear.

Weaknesses:
Major:
1. Based on my understanding the evaluation protocol is weak. Table 1 and Figure 1, are based on bootstrap consistency with with original data. Any model where the prior is stronger will perform better here. In the degenerate case, where the model does not depend on the data content (for instance the ranking is given by a fixed assignment based on data id), it seems to me that this would give Kendall Tau=1 and Top-1 agreement =1 for all datasets. Hence this evaluation protocol is not robust, as a trivial model would perform better than all the baselines.

Minor:
1. The abstract states that the BBQ achieves faster convergence which is an overstatement based on figure 3. BBQ is slower than Bayes-BT in 6 out of the 8 tasks considered.
2. The conclusion says that the method produces more accurate rankings, which from my read is not supported by the experimental evaluation.
3. The design of the experiment for Figure 4 is rather limited as well, as it only considers two items of similar skill, and it appears that there is no hetrogeneity in the rates.

---

> ### Author Rebuttal · Authors · 2026-03-31
>
> Thank you for the careful read and for noting the EM formulation and clarity.
>
> > Based on my understanding the evaluation protocol is weak. ...
>
> We agree that bootstrap consistency alone could be gamed by a data-independent ranking, which is why we also verify that the ranking recovered on the full data (our assumed reference) is sensible.
>
> Our validation is not based on bootstrap scores alone (see the third paragraph of Section 4, Experimental Results). We use two complementary checks: (i) simulation with known ground-truth order, where methods must recover the true ranking as comparisons increase, and (ii) real-data full-sample consensus checks, where rankings are compared against the full dataset and against known anchors (e.g., the reference image in IHQ).
>
> Bayes-BT and BBQ share the same Gamma prior on item skills, so the observed difference is not attributable only to stronger item-skill priors. To show that performance is not driven by one "strong" prior choice, we ablate Gamma and Beta hyperparameters on ConHa (tables below). Additional IHQ-all ablations are in our response to reviewer dKki. Top-1 agreement and Kendall's $\tau$ are not maximized at the same $(a,b)$; settings that increase Top-1 can reduce Kendall's $\tau$. These ablations and the checks described in the paper show that simply increasing prior strength does not uniformly improve all metrics, and they further support the validity of using the full-data ranking as a reference.
>
>
> **ConHa -  Bayes-BT - Top-1**
>
> | γ shape *a* \ γ rate *b* | 0.02 | 0.1 | 0.5 | 2 |
> | --- | --- | --- | --- | --- |
> | 5 | 56.30 | 56.50 | 57.70 | 55.40 |
> | 20 | 65.10 | 62.00 | 63.20 | 62.00 |
> | 100 | **72.80** | 72.00 | 70.40 | **72.80** |
>
> **ConHa -  Bayes-BT - Kendall's Tau**
>
> | γ shape *a* \ γ rate *b* | 0.02 | 0.1 | 0.5 | 2 |
> | --- | --- | --- | --- | --- |
> | 5 | 0.9072 | 0.9091 | **0.9109** | 0.9082 |
> | 20 | 0.9101 | 0.9106 | 0.9084 | 0.9081 |
> | 100 | 0.9067 | 0.9046 | 0.9000 | 0.9034 |
>
> **ConHa -  BBQ - Top-1**
>
> | γ shape *a* \ γ rate *b* | 0.02 | 0.1 | 0.5 | 2 |
> | --- | --- | --- | --- | --- |
> | 5 | 77.90 | 77.50 | **79.30** | 78.30 |
> | 20 | 76.50 | 78.50 | 77.80 | 77.00 |
> | 100 | 76.40 | 77.40 | 73.20 | 73.90 |
>
>  Beta *α* \ Beta *β* | 2 | 10 | 50 |
> | --- | --- | --- | --- |
> | 2 | 71.20 | 77.80 | 80.20 |
> | 10 | 78.80 | 80.30 | **85.90** |
> | 50 | 64.50 | 75.30 | 82.70 |
> | 200 | 58.60 | 62.00 | 76.00 |
>
> **ConHa -  BBQ - Kendall's Tau**
>
> | γ shape *a* \ γ rate *b* | 0.02 | 0.1 | 0.5 | 2 |
> | --- | --- | --- | --- | --- |
> | 5 | 0.9256 | 0.9244 | **0.9293** | 0.9271 |
> | 20 | 0.9253 | 0.9265 | 0.9267 | 0.9257 |
> | 100 | 0.9036 | 0.9111 | 0.9053 | 0.9078 |
>
> | Beta *α* \ Beta *β* | 2 | 10 | 50 |
> | --- | --- | --- | --- |
> | 2 | 0.9212 | 0.9238 | 0.8951 |
> | 10 | 0.9289 | 0.9245 | **0.9301** |
> | 50 | 0.9178 | 0.9203 | 0.9279 |
> | 200 | 0.9144 | 0.9142 | 0.9171 |
>
>
> > The abstract states that the BBQ achieves faster convergence which is an overstatement based on figure 3. ...
>
> Thank you for this correction. We will rephrase "faster convergence" precisely: BBQ improves wall-clock relative to Crowd-BT (especially at scale) and is broadly comparable to Bayes-BT, rather than uniformly faster.
>
> > The conclusion says that the method produces more accurate rankings, which from my read is not supported by the experimental evaluation. ...
>
> Agreed. Without external preference ground truth, our current metric should be described as stability/consistency relative to the full-data estimate, not absolute ranking accuracy. We will revise the conclusion wording accordingly.
>
> > The design of the experiment for Figure 4 is rather limited as well, as it only considers two items of similar skill. ...
>
> Figure 4 is a Type I error calibration experiment under the null hypothesis $H_0:\lambda_i=\lambda_j$. Equal item skill is required; if $\lambda_i \neq \lambda_j$, the setup no longer isolates Type I error and instead mixes false-positive behavior with statistical power. The two-item, equal-skill design is intentional for calibration, not meant as a full realism benchmark.
>
> > This is more of a suggestion: the evaluation protocol would be much stronger if you had access to ground truth. ...
>
> Ground-truth evaluation is ideal when available. At the same time, for topical benchmarks and large-scale crowdsourced preference datasets, ground truth is typically unavailable, which is precisely why crowdsourced comparative evaluation is used in the first place.
>
> > Can the analysis be extended to the setting where each rater ranks N items instead of pairwise? ...
>
> BBQ can be extended to the Plackett–Luce setting in the same way as other Bradley–Terry generalizations: following rank-breaking, a full ranking can be expressed as a collection of weighted implied pairwise comparisons [1]. Empirical evaluation on full ranking datasets remains future work.
>
> [1] H. A. Soufiani, D. Parkes, and L. Xia, “Computing parametric ranking models via rank-breaking,” *ICML*, 2014.

---

> > ### Author Rebuttal · Reviewer_vjFj · 2026-04-01
> >
> > While I appreciate the authors attempt to address the concerns in my review, I still think the evaluation protocol is weak, as a trivial baseline would get perfect kendall tau still. A revised set of experiments with access to ground truth data is what's needed to make the evaluation in this paper strong.

---

> > > ### Author Response · Authors · 2026-04-04
> > >
> > > We thank the reviewer for their quick and constructive response. We agree that incorporating ground-truth approximations can strengthen the evaluation, and we have updated our experiments accordingly. While absolute ground truth is not available for all datasets, several benchmarks provide strong proxies in the form of public leaderboards. These are either supersets of the pairwise comparisons or derived from independent data, and therefore serve as reasonable, model-independent references.
> > >
> > > Specifically:
> > >
> > > - **CLIC 2024**: the official leaderboard is available at [CLIC leaderboard](https://archive.compression.cc/2024/leaderboard/image_0_3/test/).
> > > - **MT-Bench**: the Arena leaderboard is available at [Arena leaderboard](https://huggingface.co/spaces/lmarena-ai/arena-leaderboard).
> > > - **HUMAINE**: the leaderboard is available at [HUMAINE leaderboard](https://huggingface.co/spaces/ProlificAI/humaine-leaderboard).
> > >
> > > ### Top1 agreement
> > >
> > > | Model | HUMAINE | MT-Bench | IHQ all | IHQ scr. | IHQ unscr. |
> > > |---|---|---|---|---|---|
> > > | Crowd-BT | 82.70 | **100.00** | 85.46 | 97.24 | 32.44 |
> > > | Bayes-BT | 97.40 | **100.00** | 75.30 | 98.89 | 23.59 |
> > > | BBQ (ours) | **98.50** | **100.00** | **99.34** | **99.86** | **61.42** |
> > >
> > > ### Kendall's tau
> > >
> > > | Model | HUMAINE | MT-Bench | IHQ all | IHQ scr. | IHQ unscr. |
> > > |---|---|---|---|---|---|
> > > | Crowd-BT | 0.8933 | **1.0000** | 0.9210 | **0.9238** | 0.8375 |
> > > | Bayes-BT | 0.9016 | **1.0000** | 0.9205 | 0.9211 | 0.8371 |
> > > | BBQ (ours) | **0.9025** | **1.0000** | **0.9211** | 0.9204 | **0.8431** |
> > >
> > >
> > > For MT-Bench, only a subset of items overlaps with the leaderboard. We therefore compute Top-1 agreement and Kendall’s Tau only for the items that exist in both the dataset and the leaderboard. This results in a saturated regime where all methods achieve perfect agreement. While not discriminative, it serves as a sanity check that all methods recover the correct ordering when sufficient signal is available.
> > >
> > > Importantly, this evaluation directly addresses Reviewer vjFj’s remaining concern that “a trivial baseline would get perfect Kendall tau.” Alignment with an external leaderboard cannot be achieved by a data-independent or trivial ranking, which would, in expectation, fail to match the independently defined ordering. The observed improvements, particularly on noisier settings such as IHQ unscrambled, are consistent with BBQ’s ability to model rater reliability and down-weight inconsistent comparisons.
> > >
> > > Overall, these results show that BBQ improves not only internal consistency but also external validity with respect to independent reference rankings. We thank the reviewer for their insightful suggestion, which motivated this addition and has strengthened the evaluation. We will include these updated experiments and discussion in the final version.

---

### Official Review · Reviewer_dKki · 2026-02-20

**Soundness:** 4
**Presentation:** 3
**Significance:** 2
**Originality:** 3
**Overall Recommendation:** 5
**Confidence:** 4

**Summary:**

The generation results of generative models are hard to evaluate since standard metrics can not reflect human preferences, while direct human labels are noisy and expensive. In order to solve this issue, this paper proposed Bayesian Bradley-Terry methods with rater Quality algorithm to get better human labels. This algorithm uses the arrival times of exponentials to quantify the probability that one result is better than another. The EM algorithm is used to estimate the parameters. The experiment results use Top-1 agreement and Kendall's Tau to evaluate the robustness and consistency of this algorithm. The result of the computation time also proves the advantage of this method.

**Compliance With Llm Reviewing Policy:**

Affirmed.

**Final Justification:**

My concerns are fully addressed. I think this paper is well-written and has solid theory and experimental support, which indicates acceptance.

**Key Questions For Authors:**

These questions are NOT weaknesses, thanks.

Q1: In rows 269-270, screened and unscreened subsets are mentioned. What are the pre-screening checks? Are they conducted with some special technique such as Brain–Machine Interface?

Q2: During one person's evaluation process, will their attention level change over the testing time? It doesn't matter if you do not have experiment result on this.

**Limitations:**

Please see the weaknesses part above.

**Strengths And Weaknesses:**

## Strengths
1. The paper is presented elegantly, making the main idea and contribution clear.

2. The algorithm is well-designed and easy to interpret with mathematical theory.

3. The related work part is good, very friendly to persons who are not familiar with BT model, and also clarifies why BT model can be applied to the pairwise comparison task.

4. The experiment parts are sufficient, able to support most of the main claims of this paper.

## Weaknesses
1. In the proposed methods, Gamma prior and Beta prior are assigned to parameters $\lambda_k$ and $q_r$. However, the choice of prior distribution seems to lack sufficient justification. Is it just a mathematical trick to help us get close-form log-likelihood? If so, do these priors really make sense?  Can we use other probability distributions, such as the uniform distribution? If so, some sensitive analysis about how different prior choices affect the results will be very helpful.

2. The idea of this new pairwise comparison method is pretty good, but the claim about how important it is is not enough. I think some more information should be provided. (1) Classic human evaluations are expensive and time-consuming. How much cost reduction this pairwise method achieve?  (2) This paper helps us get higher-quality human labels. Are these labels really necessary for model training or other tasks? If so, how much better does it work than other noisy labels?

If these two questions are adequately addressed, I would like to raise my score.

---

> ### Author Rebuttal · Authors · 2026-03-31
>
> Thank you for the positive assessment of the presentation, algorithm/theory, related work, and experiments.
>
> > In the proposed methods, Gamma prior and Beta prior are assigned to parameters λ and q. ...
>
> The Gamma prior on nonnegative item skills and the Beta prior on rater quality are chosen mainly for conjugacy, which yields closed-form EM updates and keeps inference stable and efficient. Moving away from conjugacy would require approximate inference. We agree that sensitivity to hyperparameters is important to report.
>
> Within the conjugate family, prior parameters can vary. To demonstrate robustness, we ablate the prior hyperparameters.
>
> **IHQ all -  Bayes-BT - Top-1**
>
> | γ shape *a* \ γ rate *b* | 0.02 | 0.1 | 0.5 | 2 |
> | --- | --- | --- | --- | --- |
> | 5 | 76.30 | 76.90 | 73.50 | 77.20 |
> | 20 | 91.10 | 92.60 | 92.00 | 92.10 |
> | 100 | 99.90 | 99.50 | **100.00** | 99.90 |
>
> **IHQ all -  Bayes-BT - Kendall's Tau**
>
> | γ shape *a* \ γ rate *b* | 0.02 | 0.1 | 0.5 | 2 |
> | --- | --- | --- | --- | --- |
> | 5 | 0.9237 | 0.9239 | **0.9242** | 0.9237 |
> | 20 | 0.9223 | 0.9221 | 0.9228 | 0.9219 |
> | 100 | 0.9064 | 0.9043 | 0.9050 | 0.9056 |
>
> **IHQ all -  BBQ - Top-1**
>
> | γ shape *a* \ γ rate *b* | 0.02 | 0.1 | 0.5 | 2 |
> | --- | --- | --- | --- | --- |
> | 5 | 99.40 | 99.10 | 99.70 | 98.90 |
> | 20 | 99.00 | 99.80 | 99.40 | 99.30 |
> | 100 | **100.00** | **100.00** | **100.00** | 99.80 |
>
> | Beta *α* \ Beta *β* | 2 | 10 | 50 |
> | --- | --- | --- | --- |
> | 2 | **100.00** | **100.00** | **100.00** |
> | 10 | 99.00 | 99.80 | **100.00** |
> | 50 | 90.00 | 99.20 | **100.00** |
> | 200 | 80.10 | 92.20 | 99.50 |
>
> **IHQ all -  BBQ - Kendall's Tau**
>
> | γ shape *a* \ γ rate *b* | 0.02 | 0.1 | 0.5 | 2 |
> | --- | --- | --- | --- | --- |
> | 5 | 0.9268 | **0.9275** | 0.9270 | 0.9266 |
> | 20 | 0.9223 | 0.9230 | 0.9218 | 0.9225 |
> | 100 | 0.9052 | 0.9069 | 0.9043 | 0.9054 |
>
> | Beta *α* \ Beta *β* | 2 | 10 | 50 |
> | --- | --- | --- | --- |
> | 2 | **0.9281** | 0.9216 | 0.8914 |
> | 10 | 0.9265 | 0.9268 | 0.9171 |
> | 50 | 0.9243 | 0.9260 | 0.9237 |
> | 200 | 0.9246 | 0.9245 | 0.9245 |
>
> See additional ConHa results in our response to vjFj.
>
> For all experiments in the paper, we use the same prior (Appendix B).
>
>
>
>
> > The idea of this new pairwise comparison method is pretty good, but the claim about how important it is is not enough. ...
>
> We agree that human evaluation is expensive and time-consuming, which is exactly why improving label efficiency matters. The key advantage of reliability-aware pairwise aggregation is that we can reach a trustworthy ordering with fewer comparisons and avoid collecting unnecessary additional labels.
>
> An exact percentage cost reduction is difficult to state universally because it depends on the target ranking accuracy, study design, dataset, compensation policy, and domain. Prior work argues for pairwise comparisons over MOS due to scale drift and ordering bias; for example, [1] notes that "quantifying bias on a continuous scale is even more problematic when annotators lack a common baseline." In addition, adoption of pairwise comparisons in major benchmarks (CLIC, Chatbot Arena, GDPEval [2]) supports their practical value.
>
> Higher-quality labels are also important beyond ranking itself: reducing label noise helps recover the latent ordering faster with less data, and if labels are used for training, better labels help avoid injecting incorrect supervision signals. In our experiments, this is reflected by stronger Top-1 agreement and Kendall's Tau for BBQ.
>
> > In rows 269-270, screened and unscreened subsets are mentioned. What are the pre-screening checks? ...
>
> Regarding IHQ screened versus unscreened raters: these are routine visual pre-checks on the Mabyduck platform (color-vision plates and simple contrast stimuli), not brain-machine interfaces. Full details and example figures are in Appendix D.
>
> > During one person's evaluation process, will their attention level change over the testing time? ...
>
> On attention over a session: we do not have a dedicated drift experiment. However, IHQ sessions were short (typically under about 10 minutes), so within-session drift is likely limited. We will add the average session duration and note that longer studies may exhibit stronger fatigue effects.
>
> [1] Zerman, Emin, et al. "The relation between MOS and pairwise comparisons and the importance of cross-content comparisons." Electronic Imaging 30 (2018): 1-6.
>
> [2] Patwardhan, Tejal, et al. "Gdpval: Evaluating ai model performance on real-world economically valuable tasks." arXiv preprint arXiv:2510.04374 (2025).

---

> > ### Author Rebuttal · Reviewer_dKki · 2026-04-02
> >
> > Thank you so much for your further reply. I will raise my score.

---

> > > ### Author Response · Authors · 2026-04-05
> > >
> > > We thank the reviewer for their careful reconsideration and for increasing the score following our response. We are grateful that our clarifications addressed the concerns, and we appreciate the insightful questions and suggestions, which will help us further improve the final version of the paper.

---

### Official Review · Reviewer_5Wma · 2026-03-05

**Soundness:** 2
**Presentation:** 2
**Significance:** 3
**Originality:** 3
**Overall Recommendation:** 4
**Confidence:** 3

**Summary:**

This paper introduces Bayesian Bradley-Terry model with rater Quality (BBQ), a Bayesian variant of the Bradley-Terry model that explicitly accounts for rater quality when generating rankings from pairwise comparisons. These comparisons may come from multiple raters who might be unreliable. The algorithm employs an iterative process using the expectation-maximization (EM) method at each step, ensuring consistent convergence of the likelihood function. The authors have carefully selected latent variables to achieve a closed-form solution for both the E-step (expectation) and the M-step (maximization). Empirical evidence shows that BBQ yields more robust and interpretable rankings compared to other Bayesian variants that do not consider rater quality, as well as a gradient descent-based algorithm that also explicitly models rater quality.which explicitly models rater quality.

**Compliance With Llm Reviewing Policy:**

Affirmed.

**Final Justification:**

I like the EM theory developed in the paper and most of the concerns in the paper have been addressed. I feel it deserves to be accepted in the conference.

**Key Questions For Authors:**

Please address the questions in the strengths and weakness section above.

**Technical question in Appendix A**:

In Appendix A.2, three log likelihoods are computed before computing the total data log likelihood in lines 611-634. For each of these three, the LHS of the equality (denoting whose log likelihood we compute) and the text differ. If I go by mathematical expression, I do not think the third equation is right, as it implictly assumes conditioning on $v_{r,ij}$ which is possible only when conditioned on $A$. I think the splitting of data likelihood should be as follows, instead of the one currently done.

$$\ell_c(\lambda, q; D, Z, A) = \log P(Z \mid D, A, \lambda, q) + \log P(A \mid \lambda, q) + \log P (D \mid \lambda, q, A) $$

Since the equation in line 620 is not a function of $D$, we can drop $D$ from the conditional, but the equation in line 625 should include $A$ in the conditional.

Please correct me if I missed something or said something incorrectly.


**Other questions**

1. While this paper generates scores for all items and assesses reliability for all raters, this might not be entirely necessary if the goal is simply to optimize for top-1 agreement or Kendall’s $\tau$. The authors should consider whether other algorithms—some of which were mentioned in the strengths and weaknesses box above—that explicitly compute top-k items or the best item might achieve better performance in terms of accuracy or computational efficiency. Even if it is infeasible to conduct explicit experiments to demonstrate this, I would appreciate a discussion on the topic.

Ultimately, this raises the question of how many additional parameters we estimate in relation to the objective, especially when fewer parameters may suffice.

2. In the first column of line 367, the authors say BBQ and crowd-BT perform similarly when the number of raters is small. Looking at the plot in Figure 1, it seems the authors meant bayes-BT. Could the authors clarify on this?

3. Table 2 in Appendix C omits the total number of items, which is unexpected given the raters-items interaction focus. Could you clarify if the 'number of models' column is intended to represent the total item count, or where this information is located?
4. (Minor) The authors here use $Z$ and $A$  as a latent variables. While $Z$ (sum of the lowest arrival times) has also been used as a latent variable in [1], it would be good if the authors could provide some intuition behind these choices. Are these latent variables chosen for computational reasons, i.e., to have closed-form expressions, or is there some other explanation too?

Some other presentation points:

1. While the authors use the function $l_c$ to denote conditional probability in equation (9), the authors do not explicitly define it. Althought the authors define it in the Appendix, it would be great to see a definition in the main paper too.

2. In Appendix A, I would like to thank the authors for defining all the relevant variables in a row by row format. While a careful reader can guess what the variable $D$ (that denotes the set of all comparisons) is, it would still be a good idea to remind the reader about it and define it explicitly in terms of the variables defined at the start of Appendix A. Also, shouldn't the sufficient statistics from lines 672 to 690 be conditioned on $D$ as well?

[1] Caron, F. and Doucet, A. Efficient bayesian inference for generalized bradley–terry models. Journal of Computational and Graphical Statistics, 21(1):174–196, 2012.

**Limitations:**

yes

**Strengths And Weaknesses:**

**Strengths**: The algorithm is straightforward, easy to interpret, and computationally efficient. Empirical evidence shows that it outperforms other baselines, such as the Bayesian Bradley-Terry (Bayes-BT) model, which does not account for rater quality, and the gradient descent-based Bradley-Terry model (Crowd-BT), which only focuses on maximizing the log likelihood. Additionally, the authors demonstrate that BBQ is particularly effective in scenarios involving unreliable raters.


**Weakness**:

**Soundness**: While I could verify all the theoretical claims in the paper, I could not verify the empirical results at all as code has not been provided. I would encourage the authors to provide an anonymous link to their code during the rebuttal to generate all their empirical results and the plots.

**Presentation**: I believe the authors overlooked several papers that also derive rankings from pairwise comparisons. It would be beneficial to include a comparison or discussion of these works, even if they do not utilize Bayesian inference, which is the primary focus of this paper. Given the extensive research in this field, there may be other relevant papers that have not been considered. In my opinion, the related work section mainly incorporates older studies, and some recent publications appear to have been ignored. It would be valuable to see a comparison with these newer works. The first two papers in the list below consider top-k rankings, while the last one considers full ranking.

I also think it would be good to have a slight comparison with another body of related literature that discusses personalized rankings.



1. Shah, N., Balakrishnan, S., & Wainwright, M. J. (2017). Simple, Robust and Optimal Ranking from Pairwise Comparisons.
Journal of Machine Learning Research, 18(199), 1–38.

2. Heckel, R., Simchowitz, M., Ramchandran, K., & Wainwright, M. J. (2018). Approximate Ranking from Pairwise Comparisons. In Proceedings of the 21st International Conference on Artificial Intelligence and Statistics (AISTATS), PMLR 84, 1057–1066.
3. Negahban, S., Oh, S., & Shah, D. (2017). Rank Centrality: Ranking from Pairwise Comparisons. Operations Research, 65(1), 266–287.

---

> ### Author Rebuttal · Authors · 2026-03-30
>
> Thank you for the careful and constructive review.
>
> > Soundness: While I could verify all the theoretical claims in the paper, I could not verify the empirical results at all as code has not been provided. ...
>
> Code will be released upon acceptance. We are currently working on a way to share files anonymously during rebuttal, but anonfiles is currently unavailable. The best option we have found so far is wormhole.app (link to codebase: https://wormhole.app/o4BDmJ#k-65uNI8MrFzRN4p7l8a3g), but links there expire after 24 hours. We welcome suggestions for anonymous sharing.
>
> > Presentation: I believe the authors overlooked several papers that also derive rankings from pairwise comparisons. ...
>
> Thank you for this helpful suggestion. We will cite Shah et al. (2017), Heckel et al. (2018), and Negahban et al. (2017), and clarify our positioning relative to Bayesian consensus methods with explicit rater-quality inference. Our baseline selection follows established pairwise models in GenAI benchmarks (Crowd-BT for CLIC 2018-2024 and Bayes-BT for CLIC 2025).
>
> While our paper focuses on recovering full rankings and calibrated rater qualities under a fixed comparison design, the same model can be extended to top-$k$ or best-item objectives via active pair selection and Thompson sampling [1]: comparisons can prioritize items with high posterior probability of being among the best (e.g., by sampling skills from the BBQ posterior and pairing likely winners), concentrating labeling effort on top-of-list resolution instead of distributing it uniformly across all items. This sequential, decision-focused use of the posterior is outside the scope of this paper; our experiments use a fixed design and full-ranking metrics.
>
> > Technical question in Appendix A: In Appendix A.2, three log likelihoods are computed before computing the total data log likelihood in lines 611–634. ...
>
> By definition, the complete-data log-likelihood is
> $$
> \ell_c(\lambda,q;D,Z,A)=\log P(D,Z,A\mid\lambda,q).
> $$
> The three-term expansion
> $$
> \log P(Z\mid D,A,\lambda,q)+\log P(A\mid D,\lambda,q)+\log P(D\mid\lambda,q)
> $$
> follows directly from the chain rule (equivalently $P(D,Z,A)=P(D)\,P(A\mid D)\,P(Z\mid D,A)$, with parameters in $\lambda,q$). This is a standard factorization, not an ad hoc decomposition.
>
> In the $Z$ term, the likelihood does not depend on $D$ or $q$, so we simplify to $\log P(Z\mid A,\lambda)$. In the $A$ term, it does not depend on $D$ or $\lambda$, so we simplify to $\log P(A\mid q)$. To remove ambiguity, we will revise Appendix A.2 so each conditional-independence simplification is stated explicitly next to the corresponding equation.
>
> > Other questions: While this paper generates scores for all items and assesses reliability for all raters, this might not be entirely necessary if the goal is simply to optimize for top-1 agreement or Kendall’s tau. ...
>
> We target full item skills and rater quality under a fixed design, for settings where full ranking quality matters. For use cases focused on top-$k$, extending BBQ in that direction is natural and is discussed above (active, Thompson-style use of the posterior).
>
> > In the first column of line 367, the authors say BBQ and crowd-BT perform similarly when the number of raters is small. ...
>
> Thank you for catching this: it should read Bayes-BT and BBQ (see Figure 1), and we will fix the text.
>
> > Table 2 in Appendix C omits the total number of items, which is unexpected given the raters-items interaction focus. ...
>
> Thank you for pointing this out. "Number of models" refers to number of items; we will correct the header and caption.
>
> > (Minor) The authors here use $z_i$ and $y_{ij}$ as latent variables. ...
>
> They are introduced primarily for conjugacy and closed-form EM updates, as in Caron and Doucet.
>
> > While the authors use the function $l_c$ to denote conditional probability in equation (9), the authors do not explicitly define it. ...
>
> Thank you; we will define $l_c$ explicitly in the main text.
>
> > In Appendix A, I would like to thank the authors for defining all the relevant variables in a row by row format. While a careful reader can guess what the variable $\mathcal{D}$ that denotes the set of all comparisons is, it would still be a good idea to define it explicitly in terms of the variables defined at the start of Appendix A. Also, shouldn't the sufficient statistics from lines 672 to 690 be conditioned on $\mathcal{D}$ as well? ...
>
> Thank you for the careful read. We will define $\mathcal{D}$ explicitly at the start of Appendix A.
>
> The E-step uses $\mathbb{E}[m_{r,ij} \mid \mathcal{D}, q^{(t)}, \gamma^{(t)}]$; we will make that conditioning explicit everywhere.
>
> [1] Wu, Huasen, and Xin Liu. "Double thompson sampling for dueling bandits." Advances in neural information processing systems 29 (2016).

---

> > ### Author Rebuttal · Reviewer_5Wma · 2026-04-03
> >
> > Okay regarding the technical question in Appendix A:
> >
> > I agree with the authors that the splitting they perform for $\ell_c(\lambda,q; D,Z,A)$ is correct. However, my question was on the expansion of $\log P(D \mid \lambda , q)$ in line 625.
> >
> > Given that this probability is not conditioned on A at all, how do we have $v_{r,i,j}$ in line 625? I do not think this quantity $v_{r,i,j}$ is part of dataset $D$ either. This equation only holds true when you condition on $A$ as well.
> >
> > To circumvent this problem, I had suggested the alternate splitting in the rebuttal.
> >
> > I would like to see a justification to equation (625) if you stick with this splitting mentioned in the paper.
> >
> > Rest all questions have been satisfactorily answered in the rebuttal.

---

> > > ### Author Response · Authors · 2026-04-04
> > >
> > > We are grateful for the careful read of Appendix A.2 and for the follow-up. The inconsistency you identified at lines 609 and 625 and the rest of the appendix is valid: those two displays did not match the factorization implied by the surrounding derivation. We will correct them in the revision.
> > >
> > > **Correction in the manuscript**
> > >
> > > We will update the display around line 609 (and the corresponding complete-data equation in the main text) so that the chain rule is written explicitly as
> > > $$
> > > \ell_{c}(\lambda, q ; D,Z,A) = \log P(D,Z,A\mid\lambda, q)
> > > $$
> > > $$
> > >  = \log P(Z\mid D,A,\lambda, q) + \log P(D\mid A,\lambda, q) + \log P(A\mid \lambda, q)
> > > $$
> > > $$
> > > = \log P(Z\mid A,\lambda) + \log P(D\mid A,\lambda) + \log P(A\mid q).
> > > $$
> > > We will likewise correct the display on line 625 so that its left-hand side denotes $\log P(D\mid A,\lambda)$.
> > >
> > > **Scope**
> > >
> > > The appendix already implemented the factorization you suggested in the detailed blocks; the error was limited to how the joint was written in those two places.
> > >
> > > **Summary**
> > >
> > > - The reviewer's suggested factorization for $\log P(D,Z,A\mid\lambda,q)$ is what we use in the derivation; but the lines 609 and 625 are not consistent with it.
> > > - Only those two spots are revised; the expanded final $\ell_c$ and the EM derivation (E-step and M-step) are unchanged from the submission.
> > >
> > > Thank you again for the diligence and for the constructive follow-up on line 625. We hope we have fully addressed your questions and that you are satisfied with the revision.

---

### Official Review · Reviewer_hNiM · 2026-03-11

**Soundness:** 3
**Presentation:** 3
**Significance:** 2
**Originality:** 3
**Overall Recommendation:** 4
**Confidence:** 3

**Summary:**

This paper proposes the BBQ (Bayesian Bradley-Terry with Quality) model, designed to infer item rankings from noisy pairwise comparison data while estimating the reliability of evaluators. The core method includes introducing evaluator reliability parameters into the classical Bradley-Terry model, automatically down-weighting comparisons from low-quality evaluators. The model is placed in a Bayesian framework, with item skill parameters $\lambda_i$ assigned Gamma priors and evaluator quality $q_r$ assigned Beta priors. Thurstonian latent variables $Z_{r,ij}$ are used to transform complex comparisons into tractable latent variables, and the EM algorithm is applied for iterative inference. Closed-form EM updates are provided, guaranteeing monotonic likelihood increase and convergence. The BBQ model is validated on multiple datasets (natural language, image generation, crowdsourced evaluations), demonstrating robustness and efficiency.

**Compliance With Llm Reviewing Policy:**

Affirmed.

**Final Justification:**

After reviewing the authors’ response, I feel that they have addressed my main concerns. Although I still believe that the evaluation could be further strengthened and that some of the wording could be made more precise, I no longer view these issues as serious enough to constitute fatal flaws warranting rejection. Given that the method is technically sound, reliable, and offers good interpretability, I believe the paper has reached the bar for acceptance.

**Key Questions For Authors:**

1. Is the EM + BT coupling merely a simple addition? Have you considered tighter joint modeling or nonlinear dependencies to improve adaptability to complex evaluator behavior?
2. For extremely sparse comparisons or ultra-large item/rater sets, is the numerical stability and convergence of EM updates theoretically or empirically guaranteed?
3. Have you considered extending the model to non-pairwise data (scores, rankings, multi-choice)? If so, how would the formulas and EM derivations be adjusted?
4. Is the exponential latent variable assumption robust under skewed or heavy-tailed data? Are there alternative distributional choices?

**Limitations:**

Applicable only to pairwise comparisons; does not directly handle scores, multi-choice, or ranking data. Assumes independent comparisons; sequence effects or evaluator preference interactions are not modeled.

**Strengths And Weaknesses:**

**Strengths**

- Rigorous derivations, latent variable formulation, EM update formulas, and Gamma/Beta priors are logically consistent. Convergence and monotonic likelihood increase are theoretically guaranteed.
- Automatically identifies low-quality evaluators, reducing noise impact. Robust to sparse comparisons or partially crowdsourced data.
- High computational efficiency, closed-form EM updates avoid gradient tuning and converge in seconds on the experimental datasets.

**Weaknesses**

- Method innovation is mainly a statistical improvement; the EM + BT coupling is essentially an extension of the Bradley-Terry framework and lacks core ML algorithmic novelty.
- Assumptions of independent comparisons and exponential latent variables may not hold for real crowdsourced data. A single $q_r$ may fail to reflect dynamic or context-dependent evaluator behavior.
- Experimental datasets focus on generative model evaluation, lacking validation on ultra-large item sets or extremely sparse comparisons.

---

> ### Author Rebuttal · Authors · 2026-03-31
>
> Thank you for the careful reading and for highlighting key questions around novelty, assumptions, scalability, and extensibility of BBQ.
>
> > Method innovation is mainly a statistical improvement; the EM + BT coupling is essentially an extension of the Bradley-Terry framework and lacks core ML algorithmic novelty. ...
>
> We agree that our contribution is not a new generic learning paradigm. It is a complete Bayesian inference procedure for rater quality in pairwise Bradley-Terry models, with closed-form EM updates and convergence guarantees. To the best of our knowledge, this is the first method that aggregates pairwise comparisons from noisy raters with such guarantees, explicitly models rater noise, and therefore performs better under noisy crowdsourced ratings. We position this as a principled aggregation layer for noisy human feedback, important for modern GenAI evaluation pipelines and benchmarks such as Chatbot Arena and CLIC, even when the core building blocks (BT, EM) are classical.
>
> > Assumptions of independent comparisons and exponential latent variables may not hold for real crowdsourced data. A single $q_r$ may fail to reflect dynamic or context-dependent evaluator behavior. ...
>
> Independence is a simplifying assumption, so sequence effects and interaction effects are not captured by the current likelihood. Likewise, a scalar $q_r$ captures only average rater reliability. Modeling richer behavior (context dependence, drift, interactions) would require additional structure, such as hierarchical or time-varying priors.
>
> > Experimental datasets focus on generative model evaluation, lacking validation on ultra-large item sets or extremely sparse comparisons. ...
>
> We evaluated all publicly available datasets we could find (see Table 2) that provide rater identities (i.e., which rater made which comparison) and enough annotations per rater for reliable estimation. Across these datasets, we tested settings with over 100,000 comparisons and about 2,000 raters, which is large for human-annotated benchmarks. We demonstrate performance under these conditions. We agree that this does not constitute dedicated stress testing in ultra-large $K$ or extremely sparse regimes; such stress tests could be conducted with simulated data, and we leave this for future work.
>
> > Is the EM + BT coupling merely a simple addition? Have you considered tighter joint modeling or nonlinear dependencies to improve adaptability to complex evaluator behavior? ...
>
> In our model, EM is derived on an augmented complete-data likelihood, and rater quality is inferred jointly with item skills. We agree that tighter nonlinear dependencies (e.g., item-rater interactions, time-varying $q_r$) are promising directions, but they require different likelihood constructions and likely approximate inference.
>
> > For extremely sparse comparisons or ultra-large item/rater sets, is the numerical stability and convergence of EM updates theoretically or empirically guaranteed? ...
>
> Theoretically, under standard regularity conditions, EM converges monotonically to a stationary point of the observed-data log-likelihood (each iteration increases the objective). By contrast, gradient-based training of Bradley-Terry models (e.g., Crowd-BT) depends on the learning rate: if it is too large, updates need not converge and can diverge, whereas EM has no such outer step-size tuning. These EM properties do not require a dense comparison graph.
>
> Empirically, scaling in our current implementation is dominated by memory, with comparison-structure storage of $O(K^2 R)$ for $K$ items and $R$ raters. In our experiments, settings with tens of thousands of raters and on the order of 30 items were feasible (under roughly 100 GB RAM for the settings we tested).
>
> > Have you considered extending the model to non-pairwise data (scores, rankings, multi-choice)? If so, how would the formulas and EM derivations be adjusted? ...
>
> Yes. BBQ can be extended to Plackett-Luce style ranking settings similarly to other Bradley-Terry generalizations: with rank-breaking, a full ranking can be represented as weighted implied pairwise comparisons [1]. We have not yet included empirical validation on full ranking datasets and note this as future work.
>
> > Is the exponential latent variable assumption robust under skewed or heavy-tailed data? Are there alternative distributional choices? ...
>
> The exponential/Gamma terms are auxiliary variables introduced to make the Bradley-Terry likelihood conjugate and enable closed-form EM updates [2]. They are not a direct distributional claim about annotator behavior or heavy tails in observed data. The observed data remain binary pairwise choices with rater-quality modulation.
>
>
> [1] H. A. Soufiani, D. Parkes, and L. Xia, “Computing parametric ranking models via rank-breaking,” *ICML*, 2014.
>
> [2] F. Caron and A. Doucet, “Efficient Bayesian inference for generalized Bradley–Terry models,” *Journal of Computational and Graphical Statistics*, 21(1):174–196, 2012.

---

> > ### Author Rebuttal · Reviewer_hNiM · 2026-04-03
> >
> > After reviewing the authors’ response, I feel that they have addressed my main concerns. Although I still believe that the evaluation could be further strengthened and that some of the wording could be made more precise, I no longer view these issues as serious enough to constitute fatal flaws warranting rejection. Given that the method is technically sound, reliable, and offers good interpretability, I believe the paper has reached the bar for acceptance.

---

> > > ### Author Response · Authors · 2026-04-05
> > >
> > > We thank the reviewer for their careful reconsideration of our work and for acknowledging that the concerns have been adequately addressed. We appreciate the constructive suggestions and will incorporate them to improve the final version.

---

### Decision · Program_Chairs · 2026-04-30

**Decision:**

Accept (regular)

**Comment:**

This paper proposes a Bayesian variant of the BBQ model to infer rankings from pairwise comparisons with raters of varying quality. Priors are chosen to ensure conjugacy and a (non-variational) EM algorithm is derived. The method appears to perform well when compared to a non-Bayesian model that incorporates rater quality and a Bayesian model that does not incorporate rater quality.

The major outstanding concern from reviewers was that many evaluations are based on self-consistency under the bootstrap, which could be achieved by a trivial model. The authors acknowledge this weakness but stress that ground truth rankings are not available for many datasets. They have attempted to work around this by comparing to ground truth when available and to some surrogate measures when unavailable. My judgement is that this concern is valid and I urge the authors to make it more obvious in the paper. However, I think that the surrogates are fairly convincing. Given the fundamental "reasonableness" of the underlying Bayesian model and EM algorithm, I feel comfortable recommending acceptance.